# American Stories: A Large-Scale Structured Text Dataset of Historical U.S. Newspapers

Melissa Dell[1,2*], Jacob Carlson[1], Tom Bryan[1], Emily Silcock[1], Abhishek Arora[1], Zejiang Shen[3], Luca D'Amico-Wong[1], Quan Le[4], Pablo Querubin[2,5], Leander Heldring[6]

[1]Harvard University; Cambridge, MA, USA.
[2]National Bureau of Economic Research; Cambridge, MA, USA.
[3]Massachusetts Institute of Technology; Cambridge, MA, USA.
[4]Princeton University; Princeton, NJ, USA.
[5]New York University; New York, NY, USA.
[6]Kellogg School of Management, Northwestern University, Evanston, IL, USA.
[*]Corresponding author: melissadell@fas.harvard.edu.

## Abstract

Existing full text datasets of U.S. public domain newspapers do not recognize the often complex layouts of newspaper scans, and as a result the digitized content scrambles texts from articles, headlines, captions, advertisements, and other layout regions. OCR quality can also be low. This study develops a novel, deep learning pipeline for extracting full article texts from newspaper images and applies it to the nearly 20 million scans in Library of Congress's public domain Chronicling America collection. The pipeline includes layout detection, legibility classification, custom OCR, and association of article texts spanning multiple bounding boxes. To achieve high scalability, it is built with efficient architectures designed for mobile phones. The resulting American Stories dataset provides high quality data that could be used for pre-training a large language model to achieve better understanding of historical English and historical world knowledge. The dataset could also be added to the external database of a retrieval-augmented language model to make historical information - ranging from interpretations of political events to minutiae about the lives of people's ancestors - more widely accessible. Furthermore, structured article texts facilitate using transformer-based methods for popular social science applications like topic classification, detection of reproduced content, and news story clustering. Finally, American Stories provides a massive silver quality dataset for innovating multimodal layout analysis models and other multimodal applications.

## 1 Introduction

Historical local newspapers provide a massive repository of texts about American communities and their inhabitants that can elucidate topics ranging from semantic change to political polarization to the construction of national and cultural identities to the minutiae of the daily lives of people's ancestors. Given the enormous breadth and depth of content, historical newspapers have been widely studied, yet existing open source U.S. newspaper datasets have significant limitations that complicate the extent to which modern deep learning methods can leverage and liberate their content.

Library of Congress's Chronicling America project [19] is the primary public domain historical U.S. newspaper dataset. It consists of around 20 million historical newspaper scans and their corresponding digitized texts. Its content is concentrated before 1925, as this content has entered the public

37th Conference on Neural Information Processing Systems (NeurIPS 2023) Track on Datasets and Benchmarks.

domain. Chronicling America does not recognize oftentimes complex newspaper layouts, and so digitized texts are provided at the page level, often scrambling headlines, articles, advertisements, captions, and other content regions together. Because a non-trivial share of the underlying scans are illegible, incoherent texts are prevalent, with illegibility varying across space and time. This complicates applying natural language processing (NLP) and statistical methods, and the data are not of sufficient quality to use for training a large language model to achieve a better understanding of historical English and historical world knowledge.

To address these limitations, we develop a pipeline for cheaply extracting high quality digitized article texts and layout regions from newspaper scans. First, layout detection predicts the coordinates and classes of content regions - *e.g.* articles, headlines, bylines, advertisements, pictures, etc. - using object detection methods [34]. Then, an image classifier removes illegible text bounding boxes. We next digitize the text regions using a novel optical character recognition (OCR) architecture that yields highly scalable, accurate results within our constrained budget. The focus on cost effectiveness makes the pipeline accessible to others with limited budgets who would like to digitize massive historical document collections. Finally, we associate headline, byline, and article bounding boxes. We do not process foreign language newspapers, as off-the-shelf OCR tends to perform poorly on the diverse languages and scripts.

The resulting `American Stories` (**S**tructured **t**ext **o**n **r**eporting **i**n **e**very **s**tate) dataset contains 1.14B content regions. The dataset has extensive geographic coverage across all states and has content dating as far back as the 17th century, although the bulk of content comes from the early 20th century. Figure 1 shows the distribution of scans across years and states. The vast majority of `American Stories` is older than the 72 year rule that the U.S. government uses to release personal information (*e.g.*, from the census) into the public domain.

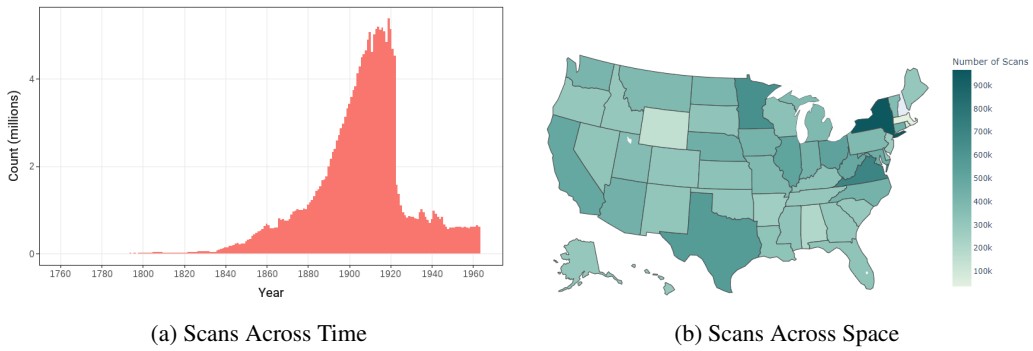

(a) Scans Across Time          (b) Scans Across Space

Figure 1: Scans in the Chronicling America database across time and space.

We show that the pipeline produces accurate predictions. The resulting texts could be used for historical language model training, or added to an external database of a retrieval augmented language model to facilitate the study of topics ranging from international events to family history. The layouts and corresponding texts could provide a massive silver quality dataset for applications like multimodal layout analysis and classification. The `American Stories` dataset also yields significantly better performance on social science analyses than the Chronicling America OCR and allows analyses that would be impossible without structured article texts. For example, we cluster article embeddings to detect which stories (e.g., Pancho Villa Expedition, 1916) received the most coverage each year.

The rest of this study is organized as follows. Section 2 discussed related literature and Section 3 describes the `American Stories` dataset. Section 4 outlines the digitization pipeline, and Section 5 evaluates the quality of the outputs. Section 6 discusses applications and Section 7 considers limitations and recommended usage.

## 2   Related Literature

Public domain newspaper datasets exist for many countries, but typically pipelines are proprietary, as the norm is to outsource digitization to a private company. Commercial newspaper databases

likewise do not disclose their pipelines, resulting in a dearth of open-source methods. The most closely related work to `American Stories` is the open-source Newspaper Navigator dataset [15]. The main, and crucial, difference between American Stories and [15] is that [15] does not detect bounding boxes of articles. [15] focuses on 7 classes of visual content: headlines, photographs, illustrations, maps, comics, editorial cartoons, and advertisements. Distinguishing articles enables legibility classification, application of custom OCR, and association of articles across bounding boxes. These tasks are at the core of our contribution, and enable the usefulness of our contribution for downstream applications. In addition, the OCR in [15] is limited to Chronicling America's OCR, which we show leads to a quality deterioration.

While we focus exclusively on texts where the entire newspaper is indisputably in the public domain (typically because it was published more than 95 years ago), it is worth noting that our pipeline could help address some of the copyright issues that have limited the public availability of historical newspapers. Outside of the nation's most widely circulated newspapers, it was rare for local papers to publish with a copyright notice or renew their copyright, required formalities until the latter half of the 20th century. Hence, the majority of local papers well into the 20th century are off-copyright. Yet these papers might sometimes print copyrighted content by third parties - *e.g.*, frequently comics, rarely ads, and occasional runs of syndicated fiction [23]. Individual news articles did not have their copyrights renewed, as copyrighting yesterday's news lacked commercial value. Detecting individual content regions - *e.g.*, so that ads and comics could be cropped out and fictional texts removed with a classifier - is a prerequisite for removing content potentially under copyright. Some copyright experts [23] have suggested this as a way forward for making historical newspapers more accessible.

## 3   Dataset

Table 1 describes `American Stories`, totaling 1.14 billion content region bounding boxes. Headlines, images, bylines, and captions are OCR'ed if legible. The dataset contains 3,313 tokens per page on average, making the full dataset 65.6 billion tokens.

| | (1) Total Boxes | (2) Articles | (3) Headlines | (4) Captions | (5) Bylines | (6) Images | (7) Ads | (8) Tables | (9) Mastheads |
|---|---|---|---|---|---|---|---|---|---|
| | | | Text Bounding Boxes | | | | Other Bounding Boxes | | |
| Legible | - | 335M | 368M | 9.7M | 14.7M | - | - | - | - |
| Illegible | - | 26M | 27M | 0.9M | 2.5M | - | - | - | - |
| Borderline | - | 77M | 22M | 1.3M | 1.2M | - | - | - | - |
| **Total** | 1.14B | 438M | 417M | 11.9M | 18.4M | 9.1M | 221M | 16.3M | 4.9M |

Table 1: `American Stories` dataset statistics.

`American Stories` provides the classes and coordinates for all content regions. Using the provided metadata, it is straightforward for users to link the coordinates with the original scans, which can be downloaded through the Library of Congress's website. We do not OCR ads because they oftentimes have unusual fonts and complex layouts, including scene text and complex tables with pricing or schedule information, complicating OCR. The table class includes tabular article data, *e.g.* sporting rosters. We do not transcribe these because accurately detecting and harmonizing the diversity of table layouts is challenging with current technology. Newspaper headers are not OCR'ed because most of their information is contained in the metadata. Mastheads - which contain subscription information - and page numbers often used very small fonts and hence are disproportionately likely to be illegible; page numbers can be inferred from the metadata.

Each text region is classified as legible, illegible, or borderline, with examples of each category shown in Figure 2. Borderline forms the grey zone between clearly legible and clearly illegible texts and is OCR'ed. Users can remove it if they would like to limit to the highest quality texts.

Substantial shares of illegible content can degrade language model training and bias downstream applications. For example, it is common for social scientists to construct data based on the presence of keyword terms [9], assigning a positive outcome if the term is present and a zero outcome otherwise. Illegibility can bias downstream analyses if it is correlated with an underlying unobserved factor. Figure 3 shows that illegibility is correlated with space and time, and hence likely to be cor-

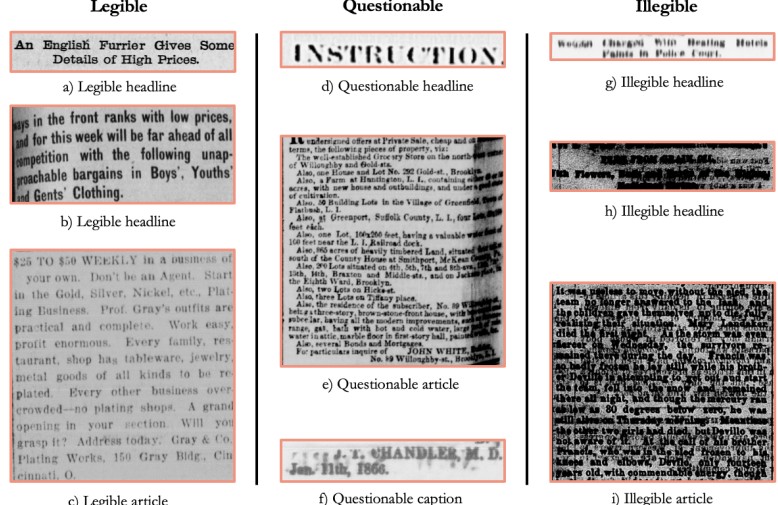

| Legible | Questionable | Illegible |
|---|---|---|
| a) Legible headline | d) Questionable headline | g) Illegible headline |
| b) Legible headline | e) Questionable article | h) Illegible headline |
| c) Legible article | f) Questionable caption | i) Illegible article |

Figure 2: Examples of legibility classification, as predicted by our trained model.

related with unobserved factors. Using our data, researchers can remove papers with high illegibility rates if desired and more realistically assess selection into the database.

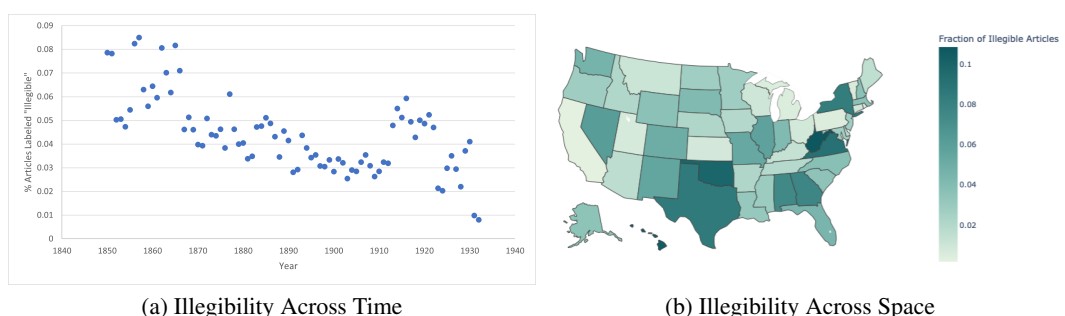

(a) Illegibility Across Time      (b) Illegibility Across Space

Figure 3: Illegible articles in the Chronicling America database across time and space.

`American Stories` has a Creative Commons CC-BY license, to encourage widespread use. The data are available on the Hugging Face Hub[1]. The raw files are in a json format, and the Hugging Face repo comes with a setup script that easily allows people to download both raw and parsed data to facilitate language modeling and computational social science applications. The supplementary materials and the Readme on the dataset repository provide a detailed usage guide.

## 4 Methods

**Overview:** The `American Stories` pipeline consists of four steps: layout/line detection, legibility classification, OCR, and article association. The pipeline is available on Github[2].

The pipeline's modularity offers several advantages. Theoretically, localization (of layouts, lines, words, and characters) and recognition (of characters, akin to classification) may rely on different features of the image, suggesting modularity [31]. Practically, there are vast differences in the number of labels required for training each component of the pipeline. Modularity also leads to architectural simplicity. Swapping in different encoders is straightforward, which makes the pipeline more customizable and future-proof. Off-the-shelf models performing a single pipeline task - like Segment Anything [14] or an OCR engine - would be straightforward to swap in.

---

[1] https://huggingface.co/datasets/dell-research-harvard/AmericanStories
[2] https://github.com/dell-research-harvard/AmericanStories

The `American Stories` pipeline uses architectures designed for mobile phones - Yolo v8 [34] and MobileNet v3 [10] - because the accuracy hit relative to much larger models was very modest and deployment costs were over an order of magnitude lower. If budget is not a concern, it would be straightforward to swap in a vision transformer, (*e.g.*, [4, 1, 21]) and a two stage object detection framework (*e.g.*, [2]), variations that [3] examine quantitatively on Chronicling America.

The pipeline was run on Azure F-series CPU nodes. Training used an Nvidia A6000 GPU card. Details are described in the Supplementary Materials.

**Layout and Line Detection:** Layout region coordinates and classes are detected using Yolo v8 [34], with Figure 4 showing examples. We train the layout detection model on 2,202 labeled newspaper images, consisting of 48,874 layout objects, with an average of 22 layout objects per page. All annotations for the pipeline were created by undergraduate research assistants, with scans selected randomly and using active learning [27]. To develop a general purpose model, we annotated scans of public domain and off-copyright newspapers from throughout the 19th and 20th centuries. Scans from Chronicling America comprise 13% of the training sample.

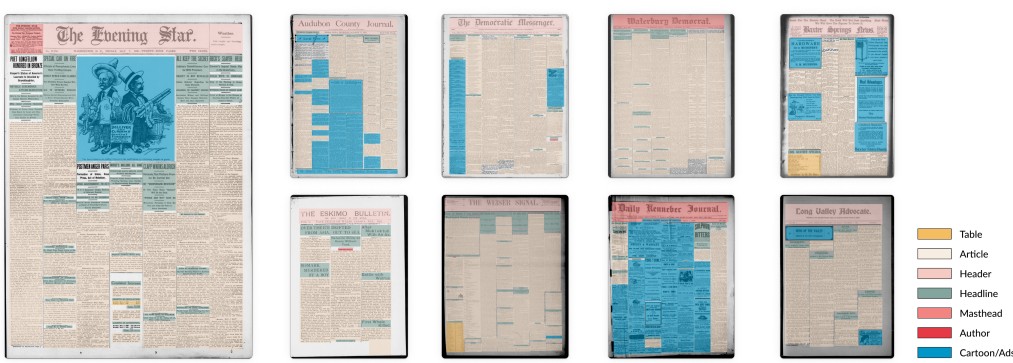

Figure 4: Variety of newspaper layouts with our layout detection pipeline outputs overlaid.

For content regions that are OCR'ed, we detect individual lines within each region using Yolo v8, as lines are the input to OCR. The model is trained on 4,000 synthetic and 373 annotated line crops. Yolo v8, like other object detection frameworks, takes square images as inputs. When the aspect ratios of content regions differ significantly from squares, downstream OCR performance tends to be adversely affected. Hence, for content regions with an aspect ratio greater than 2:1 (twice as tall as wide), we split the layout region (with overlap), run line detection over each separate box, and then run non-maximum suppression jointly over the resulting predictions. For the same reason, before sending lines to OCR, we split lines with an aspect ratio below 1:30 (thirty times wider than tall).

**Legibility Classification:** We classify headline, article, byline, and caption regions as legible, illegible, or borderline (see Figure 2) using an image classifier with a Mobilenet v3 backbone that is trained on 1,094 double-labeled content region crops.

**OCR:** We aimed to deploy a highly accurate digitization pipeline within a constrained cloud compute budget of $60,000 USD. As documented in detail in the supplementary materials, existing OCR solutions did not meet these requirements. Commercial solutions were far too costly, and TrOCR Base [18] - an open-source transformer sequence-to-sequence OCR that produced accurate results - was nearly fifty times more costly to deploy on cloud CPUs than our budget. More scaleable open-source OCR engines were noisy even when fine-tuned on newspaper annotations.

We met our accuracy and cost objectives with the EfficientOCR framework [3]. EfficientOCR uses deep learning-based object detection methods to localize individual characters and/or words in an image. Character/word recognition is modeled as a character/word-level image retrieval problem, using a vision encoder contrastively trained on character/word crops, largely created through augmenting digital fonts. At inference time, character/word embeddings are decoded to text in parallel by retrieving their nearest neighbor from an offline index of exemplar character or word embeddings, created by rendering character/word images with a digital font. Distances are computed using cosine similarity with a Facebook Artificial Intelligence Similarly Search (FAISS) backend [11].

Switching between character and word recognition is important. There are many terms that would not appear in even a very large dictionary, as narrow newspaper columns imply that hyphenated words at the end of lines are common.[3] The texts also have a long-tailed distribution of proper nouns and antiquated acronyms and words. Hence, at inference time, whenever a word crop is below a tuned cosine similarity threshold of 0.82 from its nearest neighbor in the offline word embedding index, we instead apply character-level EfficientOCR to individual character crops within the word. The supplementary materials provide a detailed description of training and deployment.

**Content Association**: We associate headlines together (if spanning multiple boxes), associate them with bylines (if present), and with the first article bounding box, using rule-based methods that exploit the position of article and byline bounding boxes relative to headlines (see the Supplementary Materials). Rule-based methods perform less well for articles spanning multiple columns or pages, and language understanding is required. However, we find on a labeled sample that only around 3.8% of articles span multiple article bounding boxes, and only 0.2% of articles span multiple pages, meaning there is little scope for gains from neural methods. (Articles spanning multiple columns and pages become more common after the Chronicling America period, as the price of paper falls and font size increases.) Since these cases are rare and our compute budget is limited, we do not associate multiple article bounding boxes together for this release, but may do so in the future, using the RoBERTa cross-encoder method developed in [29].

## 5   Pipeline Evaluation

**Measurement:** We use four carefully constructed datasets to evaluate the pipeline:

- **Full page scans:** Two student annotators labeled layout regions and hand-entered the texts for 10 full page scans, resolving all discrepancies by hand. The set consists of 597 content regions and 196,655 characters. It allows us to evaluate the end-to-end pipeline, which requires transcribed full-page scans since layout analysis is applied at the page level. We further use this sample for evaluating content association. It contains 214 headline-article bounding box pairs.

- **Transcribed day per decade sample:** To evaluate line detection and OCR on highly diverse content, we hand-transcribe a randomly selected sample of 50 lines for each decade between 1850-1920. During this period, printing and archival technology changes significantly. Examples of these textlines, with their accompanied EffOCR transcriptions, are shown in the supplemental materials as Table 1.

- **Transcriptions of randomly selected lines from Carlson et al. [3]**: This sample is used to report comparisons to other object detection frameworks, backbones, and OCR engines. These comparisons are taken from [3], which develops EfficientOCR. This sample includes 64 textlines drawn randomly from random scans in the Chronicling America collection.

- **Legibility sample**: We evaluate legibility on a randomly selected (from legibility training data), double labeled sample of 100 bounding boxes, 50 headlines and 50 articles. Transcriptions are not included, as we cannot create character labels for illegible content.

We measure pipeline accuracy with the character error rate (CER), defined as the Levenshtein distance [16] between the end-to-end digitized content and the ground truth, divided by the length of the ground truth. OCR is similarly evaluated by the character error rate on ground truth layout and line annotations (and hence does not include transcriptions errors induced by errors in the layout predictions). We also examine non-word rate, as it does not require costly-to-create labeled data so can be evaluated on a much larger sample, though it is more difficult to interpret. To measure

---

[3]Our word dictionary consists of words rendered three times each: with all lowercase characters, all uppercase characters (common in headlines), and capitalized. We select words by first taking the top 25,000 words from a modern dictionary that ranks word frequency [5]. We remove 3,999 words that never appear in a sample of all Chronicling America scans from one-day-per decade (1850s-1920s), OCR'ed with character EfficientOCR. Inspection revealed that these were overwhelmingly words related to modern concepts like computing and modern medicine. We then added the 500 words that most frequently appear in this sample but were not frequent modern words - mostly consisting of terms from antiquated domains like traditional medicine - and also added numbers, contractions, state and month abbreviations, and punctuation, for a total of 22,230 total terms.

the quality of layout and line detection, we use mean average precision (mAP@50:95), as well as decomposing what share of the overall character error rate is due to layout and line detection errors. For full article association, we focus on confusability between legible and illegible scans. Finally, we evaluate content association using F1.

**Overall Pipeline Evaluation:** The end-to-end CER is 0.051 (Table 2), showing the pipeline's high overall accuracy. Some of the errors - *e.g.* confusing commas and periods are particularly prevalent - are straightforward to fix in post-processing. When we apply a lightweight spellchecker [5], the CER falls to 0.044. Spellchecking produces a slight increase in CER in headlines, likely due to a higher concentration of proper nouns.

|  | (1) Overall | (2) Headlines | (3) Articles | (4) Ads |
|---|---|---|---|---|
| Mean Average Precision | 63.48 | 88.64 | 91.31 | 78.4 |
| Overall CER (Spellchecked) | .044 | .092 | .038 | - |
| Overall CER | **.051** | .089 | .049 | - |
| CER from OCR | .043 | .071 | .039 | - |
| CER from layout detection | .012 | .018 | .010 | - |
| Article Association F1 | 97.0 | - | - | - |

Table 2: Pipeline evaluation on ten labeled scans. CER is the character error rate, decomposed into errors from OCR and from layout detection. Article association F1 evaluates the association of headlines with each other and the first article bounding box. Spellchecking is applied after EffOCR, using [5].

Figure 5 plots the non-word rate at the scan level on a day-per-decade sample, where non-word rate measures the share of terms not in a lengthy dictionary with 82,765 terms [5]. Even with a perfect OCR, the non-word rate could be appreciable, due to hyphenated words at the end of rows, acronyms, proper nouns, and antiquated terms. The `American Stories` distribution is concentrated well to the left of Chronicling America's distribution, underscoring the quality of our texts. Differences in the non-word rate between Chronicling America and American Stories are likely to reflect both differences in OCR quality and differences in filtering content based on legibility and content region type, as Chronicling America digitizes all the texts it can localize on the page.

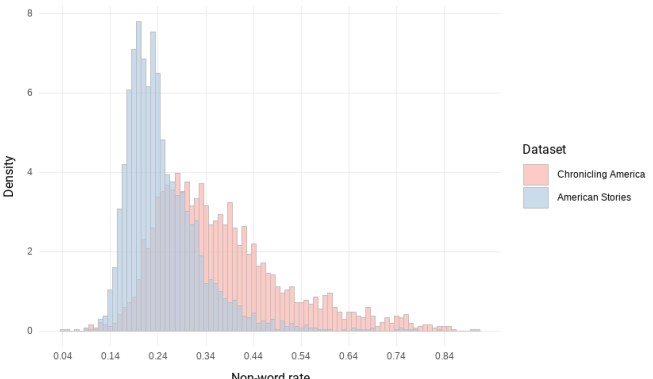

Figure 5: Non-Word Rate Distributions of `American Stories` and Chronicling America.

**Layout and Line Detection Evaluation:** mAP, reported in Table 2, is high, particularly for the classes of central interest such as headlines (88.64) and articles (91.31). Confusing ads for articles - as they often look similar in this period - is relatively common. The ads that look like articles tend to OCR well and have natural language texts, so these errors are also tolerable. The overwhelming majority of content regions in the ten labeled scans are articles, headlines, and advertisements, as photographs - a rarity in this period - do not appear. The mAP for line detection is 86.20.

We also decompose the overall CER into errors due to OCR and errors due to line and layout detection. When the layout model fails to detect text regions, misclassifies them as another category such as ads that isn't OCR'ed, or when line detection misses or crops lines, this increases the CER. The CER due to layout and line detection errors is 0.012.

**Legibility Classification Evaluation:** We want to avoid classifying legible texts as illegible and vice versa, with the borderline class existing to encompass the grey zone between these two categories. In a random sample of labeled texts - containing 81 legible texts - none of the legible texts are misclassified as illegible. Of 16 illegible texts, only one is misclassified as legible. To further evaluate this, we ran all content regions from a one-day-per-decade sample through OCR. The non-word rate is more than three times higher for illegibly classified content as compared to legibly classified content, and more than twice as high for illegibly classified content as for borderline content.

**OCR Evaluation:** Table 2 reports the CER from running OCR on ground truth layouts and lines. It is 0.043. The supplementary materials provide a much more detailed analysis of OCR quality. Evaluation on a day per decade sample shows that CER ranges from 8.9% (1850s) to 1.8% (1910s), with the bulk of content concentrated in the later period where scans are less challenging. The supplementary materials also document that EffOCR [3] best meets our accuracy and cost requirements, by comparing to a variety of open-source and proprietary OCR engines.

**Content Association Evaluation:** We achieve an F1 of 97 for associating headlines with articles on our labeled evaluation dataset. We associate all bylines correctly.

# 6   Applications

**Language Modeling**: `American Stories` provides a massive amount of text that can be used to continue pre-training large transformer language models, helping a language model to develop a better understanding of 19th and early 20th century English and greater world knowledge about the past. Moreover, a retrieval augmented language model (e.g. [12]), combined with longer context windows, could provide a valuable tool for retrieving and summarizing vast information, whether it be perspectives on paradigm-shifting historical events or the minutiae of the daily lives of people's ancestors. `American Stories` also provides extensive content for studying semantic change.

**Multimodal Classification**: The layouts and texts in `American Stories` comprise a rich multi-modal dataset, providing vast silver quality data that could be used for developing novel methods for multimodal layout analysis or multimodal classification - *e.g.*, with image-caption pairs.

**Topic Classification:** Topic classification of texts in historical newspapers is a common social science application, with the literature overwhelmingly using keyword searches to measure topics [9]. To evaluate how `American Stories` can facilitate topic classification, relative to the existing Chronicling America page-level OCR, we use neural and sparse methods to classify whether a randomly selected set of content is about politics, a frequently covered topic. We use a RoBERTa large [20] classifier, applied to articles in `American Stories` and chunks of Chronicling America (the page OCR is significantly longer than the context window). The training data were randomly sampled at the article level and contain 2418 articles. The development set contains 15 randomly selected full scans (498 articles) and the test set contains 62 randomly selected scans (1473 articles). We also consider keywords, using two different approaches for selecting these: keyword mining on the training set and asking ChatGPT, with careful prompting. All methods are described in the supplementary materials.

`American Stories` supports article level classification, whereas Chronicling America only supports page level classification. A page is a positive example in the ground truth if any of the articles are on topic, and article or chunk predictions can be aggregated to page level predictions using the same definition. Retrieval at the scan level tends to retrieve a lot of extraneous content, as typically only part of the page contains articles about politics.

On structured article texts, neural methods outperform keyword methods by a wide margin (F1 of 83.6 versus 58.1). All methods perform better at the page level, as there is a much lower chance of false negatives, with the best performance by a wide margin coming from applying neural methods to the `American Stories` corpus.

**Content Dissemination Networks:** Reproduced content is of considerable interest to social scientists [7], but detecting it can be challenging due to OCR noise and abridgement. We evaluate this task on a full-day sample of labeled front pages from March 1, 1916, a random day that consists of 113 reproduced articles (the median article is reproduced twice) as well as 1,994 singleton articles.

| | Topic classification | | | Reproduced content | |
|---|---|---|---|---|---|
| | Neural F1 (1) | Sparse Mining (2) | GPT (3) | Neural ARI (4) | Sparse Viral Texts (5) |
| *Article Level* | | | | | |
| American Stories | **83.6** | 56.4 | 58.1 | **75.1** | 32.3 |
| *Page Level* | | | | | |
| American Stories | **96.0** | 82.8 | 79.6 | **86.2** | 74.6 |
| Chronicling America | 83.3 | 83.7 | 79.6 | - | 71.4 |

Table 3: Comparing American Stories to Library of Congress's Chronicling America.

To detect reproduced content at the article level with `American Stories`, we deploy the pre-trained neural model from [28]. They contrastively tuned a Sentence BERT model [26, 32] on a large, hand-annotated sample of paired reproduced articles from a later period. At inference time, article representations are clustered using single linkage clustering to detect reproduced content. To make the `American Stories` result comparable with Chronicling America, we amalgamate the predictions to the page level. A page-pair is counted as positive if the pages have any article in common, making the task easier.

For detecting reproduced content with the Chronicling America full page scans, where we lack the article texts required for the neural method, we deploy the sparse methods from Viral Texts [30]. Viral Texts was designed specifically for detecting reproduced texts in Chronicling America's noisy page-level OCR by looking for overlapping $n$-gram spans. We also apply Viral texts to `American Stories` at the article and page level. Table 3 reports the adjusted rand index (ARI) for these specifications. The Viral Texts method gives slightly better results on `American Stories`, but the real advantage of `American Stories` is that the article structure allows the use of a neural method, which dominates the sparse method by nearly 12 percentage points at the page level.

**News Story Clustering:** The structured nature of `American Stories` allows for further article-level clustering of content. As a demonstration of this, we show how articles can be grouped into news stories, with different articles that are part of the same unfolding news story clustering together. This prediction is not possible with the unstructured page content in Chronicling America.

| Year | Biggest story | Year | Biggest story |
|---|---|---|---|
| 1885 | Death of General Grant | 1903 | Panama Canal Treaty |
| 1886 | Southwest Railroad Strike | 1904 | Russo-Japanese War |
| 1887 | Vatican supports Knights of Labor | 1905 | Russo-Japanese Peace Process |
| 1888 | Rail strikes | 1906 | Hepburn Railroad Rate Bill |
| 1889 | Samoan Crisis | 1907 | Mining accidents |
| 1890 | 1893 World's Fair planning | 1908 | Taft presidential victory |
| 1891 | New Orleans Lynchings | 1909 | Race to the North Pole |
| 1892 | Homestead Steel Strike | 1910 | Rail strikes |
| 1893 | World's Fair, Chicago | 1911 | Canadian Reciprocity Bill |
| 1894 | Wilson–Gorman Tariff Act | 1912 | Republican National Convention (Taft v Roosevelt) |
| 1895 | British occupation of Corinto, Nicaragua | 1913 | Underwood-Simmons Tariff Act |
| 1896 | Bimetallism Movement | 1914 | World War I |
| 1897 | Coal Miners' Strike | 1915 | World War I |
| 1898 | Cuban War of Independence | 1916 | Pancho Villa Expedition |
| 1899 | Philippine-American War | 1917 | World War I |
| 1900 | Anglo-Boer War | 1918 | World War I |
| 1901 | U.S. Steel Recognition Strike | 1919 | Treaty of Versailles |
| 1902 | Anthracite Coal Strike | 1920 | Rail strikes |

Table 4: Largest news story in each year, 1885-1920.

To create these clusters, we fine tune a contrastive biencoder on modern news texts, scraped from `allsides.com`, a website which amalgamates different representations of the same news story from different news outlets. Full details of the data, model and training are given in the supplementary materials. We ran this model over all de-duplicated front page articles from 1885-1920. We read 20 random articles in the largest cluster for each year and named the story which the articles in the cluster refer to. Table 4 shows the largest news story cluster by year.

This application demonstrates one of the many ways that the structured article texts in `American Stories` can be used to unlock new ways of studying historical and social questions.

In addition to these tasks, image-caption pairs from layout analysis could be used for training and assessing image captioning models, visual question-answering models, cross-modal retrieval models, image retrieval models, and multi-modal understanding (e.g., representation learning) models. `American Stories` could also be leveraged to substantially lower the costs of creating new benchmark datasets for other common ML tasks, e.g., image and text classification, image retrieval, and named entity recognition.

## 7   Limitations and Recommended Usage

`American Stories` contains historical language, that reflects the semantics and cultural biases of the time. This is a distinguishing feature, that is core to many potential applications. We do not filter texts that use antiquated terms or that may be considered offensive, as this would invalidate the use of the dataset for studying semantic change and historical contexts. At the same time, this makes `American Stories` less suited for tasks that require texts that fully conform to current cultural standards or semantic norms. For these reasons, we recommend against the use of `American Stories` for training generative models. While the OCR is high quality, `American Stories` is also not well-suited to tasks requiring fully clean texts. Rather, `American Stories` can be used for a wide variety of applications, ranging from elucidating social science questions to training a historically-oriented language model to exploring world and family history. It also provides a modular pipeline that can be customized for other document collections and scaled cheaply, offering a blueprint for liberating large-scale historical text corpora.

## Acknowledgement

Funding was provided by the Harvard Data Science Initiative, Harvard Catalyst, the Ken Griffin Harvard Economics Fund, and Microsoft Azure compute credits. We thank Anoushka Ashwin, Domenick Clark Regina, Chloe Combes, Will Cox, Connor Fogal, Brevin Franklin, Prabhav Kamojjhala, Zachary Lee, Roberto Lopez-Irrisarry, Elan Pelegri, Krishna Prasad Srinivasan, and Sarah Strohecker for research assistance.

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

# Supplementary Materials

## Model Details

`American Stories` deploys a modular pipeline to digitize historical newspapers. This section provides details for each component of the pipeline.

### Layout Detection

To detect articles, headlines, ads, and other content regions in a newspaper scan, we deploy YOLOv8 (Medium) [34], initialized from the officially released YOLOv8m pretrained checkpoint. We train for 100 epochs on 2,202 labeled newspaper scans with 48,874 total layout objects, using default YOLOv8 hyperparameters except: `{imgsz: 1280, iou: 0.2, max_det: 500}`. The final model achieved a 0.91 mAP50:95 on article bounding boxes and a 0.84 mAP50:95 on headline bounding boxes. We decreased the confidence threshold to 0.1 to increase article and headline recall.

### Legibility Classification

Text image bounding boxes are classified as legible, borderline, or illegible, using MobileNetV3 (Small) [10] initialized from the PyTorch Image Models ("timm") [35] pretrained checkpoint. We train for 50 epochs on 979 labeled article, headline, and image caption examples. 678 of the labeled examples were legible, 192 borderline, and 109 illegible. The model was trained with weighted Cross Entropy Loss: weights [2.0, 1.0, 1.0] for legible, borderline, and illegible classes, respectively. The following specifications were used: `{resolution: 256, learning rate: 2e-3}`. The learning rate was multiplied by 0.1 every twenty epochs.

### Text Line Detection

Line bounding boxes are detected using YOLOv8 (Small) [34] initialized from the official YOLOv8s pretrained checkpoint. We train first for 100 epochs on 4000 synthetically generated articles, with default YOLOv8 hyperparameters. After synthetic training, the model was additionally trained for 50 epochs on 373 hand-annotated article and headline crops, with default YOLOv8 hyperparameters except for the following: `{resolution: 640, initial learning rate: 0.02, final learning rate: 0.002}`.

### Word and Character Localization

Words and characters are detected using YOLOv8 (Small) [34], initialized from the official YOLOv8s pretrained checkpoint. We train first for 100 epochs on 8000 synthetically generated textlines with default YOLOv8 hyperparameters. After synthetic training, the model was additionally trained for 100 epochs on 684 hand-annotated text line images, with default hyperparameters except for the following: `{resolution: 640, initial learning rate: 0.02, final learning rate: 0.001}`. Each hand-annotated line image was replicated three times with random augmentations along three axes: random rotation between -1°and 1°, random image brightness shift from -30 to 30%, and randomly applied blur at the 0-4px level. On average, text line examples contained 4.3 words and 23.6 characters.

### Word Recognition

Building upon the architecture in [3], we train word recognition as a nearest neighbor image retrieval problem. As described in the main text, the training dataset for the model consists of digital renders of words created using 43 fonts, silver quality data from the target dataset created by applying the EffOCR-C (Small) model from [3] to a random sample of days, and a small number of randomly selected hand labeled word crops. We limited the number of crops with model-generated labels to 20 - so each word can have 0-20 silver-quality crops depending upon its frequency of occurrence in our random sample. This limit is binding for common words, *e.g.,* "the".

The recognizer is trained using the Supervised Contrastive ("SupCon") loss function [13], a generalization of the InfoNCE loss [24] that allows for multiple positive and negative pairs for a given anchor. In particular, we work with the "outside" SupCon loss formulation

$$\mathcal{L}_{\text{out}}^{\text{sup}} = \sum_{i \in I} \mathcal{L}_{\text{out},i}^{\text{sup}} = \sum_{i \in I} \frac{-1}{|P(i)|} \sum_{p \in P(i)} \log \frac{\exp\left(\mathbf{z}_i \cdot \mathbf{z}_p / \tau\right)}{\sum_{a \in A(i)} \exp\left(\mathbf{z}_i \cdot \mathbf{z}_a / \tau\right)}$$

as implemented in PyTorch Metric Learning [22], where $\tau$ is a temperature parameter, $i$ indexes a sample in a "multiviewed" batch (in this case multiple fonts/augmentations of the same word), $P(i)$ is the set of indices of all positives in the multiviewed batch that are distinct from $i$, $A(i)$ is the set of all indices excluding $i$, and $z$ is an embedding of a sample in the batch [13].

To create training batches for the recognizer, we use a custom $m$ per class sampling algorithm without replacement, adapted from the PyTorch Metric Learning repository [22]. The $m$ word variants for each class (word) are drawn from both target documents and augmented digital fonts. We select $m = 4$ and the batch size is 1024, meaning 4 styles of each of 256 different words appear in each batch. For training without hard negatives, we define an epoch as letting the model see each word (case-sensitive) exactly $m = 4$ times. Sampling for each class occurs without replacement until all variants are exhausted.

In order to converge faster with limited compute, we also implement offline-hard negative mining, batching similar negatives and their corresponding positive anchors together - thus making the contrasts between the positive and negative pairs within a batch especially informative. To create hard negative sets, we render each word using a reference font (Noto-Serif Regular) and embed it to create a reference index. We find $k = 8$ nearest neighbors for each word using this index and the model trained without hard negatives, which yields sets of 8 words that have a similar appearance when rendered with the reference font. We use only the reference font to create these sets because using crops corresponding to all 43 fonts for each word is computationally costly and creates more hard negative sets than we can use in training. We also use each word crop from the target dataset (both silver quality annotations generated with model predictions and gold quality human-annotated predictions) to create hard negative sets. Hence, the total number of hard-negative sets equals the number of words in our dictionary (generated with the reference font) plus the number of word crops from the newspaper data in the training set.

Each hard negative set contains 8 words, with $m = 4$ views per word, which means we can fit 32 randomly sampled hard negative sets within each batch. An epoch is defined as seeing each hard negative set once. Since the number of synthetic views of an image is much larger than the number of target newspaper crops, whenever newspaper crops are available we force the $m$ views of a word to contain an equal number of synthetic and target crops.

We use a MobileNetV3 (Small) encoder pre-trained on ImageNet1k sourced from the timm [35] library, more specifically, the model *mobilenetv3_small_050*. We use 0.1 as the temperature for SupCon loss and AdamW as the optimizer with Pytorch [25] defaults for all parameters other than weight decay (5e-4) and learning rate. We used Cosine Annealing with Warm Restarts as the learning rate scheduler with a maximum learning rate of $2e - 3$, a minimum learning rate of $0$, time to first restart ($T_0$) as the number of batches in an epoch, and restart factor, $T_{mult}$ of 2 using the implementation provided in Pytorch.

While fonts and newspaper crops for each word act as an augmentation on the skeleton of the word, we also add more image-level transformations to improve generalization. These include Affine transformation (only slight translation and scaling allowed), Random Color Jitter, Random Autocontrast, Random Gaussian Blurring, Random Grayscale, Random Solarize, Random Sharpness, Random Invert, Random Equalize, Random Posterize and Randomly erasing a small number of pixels of the image. Additionally, we pad the word to make the image square while preserving the aspect ratio of the word render. We do not use common augmentations like Random Cropping or Center Cropping, to avoid destroying too much information.

The model trained without hard negatives was trained for 50 epochs and with hard negatives, it was trained for 40 epochs. For selecting the best checkpoint, we use 1-CER (OCR Character Error Rate) as the validation metric on the validation set from [3]. We chose the model that performed best in terms of CER when detecting only words on the validation set. This means that if a word is outside

of our dictionary, it is forcefully matched to the nearest neighbor in the dictionary. The best model achieved a CER of 4.9% with word-only recognition.

At inference time, words are recognized by retrieving their nearest neighbor from the offline embedding index created with the reference font, using a Facebook Artificial Intelligence Similarity Search backend [11]. The code to train the model and generate training data, as well as the model checkpoints, are made available on our GitHub repo.[4]

### Character Recognition

When the nearest neighbor to an embedded word crop in the offline word embedding index is below a cosine similarity threshold of 0.82, we default to character-level recognition. We use the EffOCR-C (Small) model that is developed in [3] for character recognition.

### Content Association

This step associates headlines, bylines, and article bounding boxes. We use rule-based methods that exploit the position of article and byline bounding boxes relative to headlines. Specifically, we associated a headline bounding box with an article bounding box if they overlap vertically by more than 1% of the page width, and the bottom of the headline is no greater than 10% of the page height above the top of the article, and no greater than 2% of the page height below the top of the article. If multiple article bounding boxes satisfy these rules for a given headline, then we take the highest. The same rules are used to associate bylines.

## Pipeline Evaluation

As discussed in the main paper, we evaluate the data processing pipeline in an end-to-end fashion, as well as evaluating individual sections, particularly OCR. Here we provide additional details on those evaluations.

### OCR Evaluation

Processing 20 million scans required a cost-effective OCR solution, and downstream tasks require highly accurate OCR. We compared custom, open-source, and commercial OCR solutions by accuracy, speed, and cost to determine our final architecture. Character Error Rate measurements were made on two separate validation datasets:

- **CER [3]** Error rate on a dataset of 64 randomly selected Chronicling America textlines, sampled from the entire collection. Textlines were randomly sampled from random scans, then cropped and transcribed. This dataset and its construction is described in detail in [3].

- **CER Day-Per-Decade** Error rate on a sample of 225 total textlines, sampled from all scans in the Chronicling America collection published on March 1st of years ending in "6," from 1856-1926. Unlike the above sample from [3], this sample is balanced across the time periods the predominate the Chronicling America collection. 25 textlines were sampled randomly from random pages published on each of the days. A selection of textlines from this collection, along with their EffOCR transcriptions, are shown in Figure S-1. This dataset is designed to be much more challenging than the first, weighting older, harder to read scans more heavily despite their relative scarcity in the Chronicling America collection.

Comparisons are listed in Table S-1. Training procedures for EffOCR-Word are described above. See [3] for training procedures, initialization checkpoints, and additional details on training and evaluating comparison models.

Of the options we examined, EffOCR-Word (Small) was the clear best option, providing a Character Error Rate under 5% on the hardest evaluation set while offering the cheapest rate per line on an Microsoft Azure Fs4v2 instance.

---

[4] https://github.com/dell-research-harvard/AmericanStories.

| Model/Engine | Seq2Seq? | Transformer? | Pretraining | Parameters | CER [3] | CER Day-Per-Decade | Lines Per Second | Cost Per 10K Lines |
|---|---|---|---|---|---|---|---|---|
| EffOCR-C (Base) | × | × | from scratch | 112.5 M | 0.023 | 0.062 | 0.27 | $1.77 |
| EffOCR-C (Small) | × | × | from scratch | 9.3 M | 0.028 | 0.080 | 7.28 | $0.06 |
| EffOCR-T (Base) | × | ✓ | from scratch | 101.8 M | 0.022 | 0.059 | 0.17 | $2.80 |
| **EffOCR-Word (Small)** | × | × | **from scratch** | **10.6 M** | **0.015** | **0.043** | **11.60** | **$0.04** |
| Google Cloud Vision OCR | ? | ? | off-the-shelf | ? | 0.005 | 0.019 | ? | $15.00 |
| Tesseract OCR (Best) | ✓ | × | off-the-shelf | 1.4 M | 0.106 | 0.170 | 2.43 | $0.19 |
| EasyOCR CRNN | ✓ | × | off-the-shelf | 3.8 M | 0.170 | 0.274 | 10.75 | $0.04 |
| | | | fine-tuned | | 0.036 | 0.157 | | |
| | | | from scratch | | 0.131 | - | | |
| PaddleOCR SVTR | × | × | off-the-shelf | 11 M | 0.304 | | 7.36 | $0.06 |
| | | | fine-tuned | | 0.103 | | | |
| | | | from scratch | | 0.104 | | | |
| TrOCR (Base) | ✓ | ✓ | off-the-shelf | 334 M | 0.015 | 0.038 | 0.23 | $2.02 |
| | | | fine-tuned | | 0.013 | 0.027 | | |
| | | | from scratch | | 0.809 | - | | |
| TrOCR (Small) | ✓ | ✓ | off-the-shelf | 62 M | 0.039 | 0.121 | 0.53 | $0.90 |
| | | | fine-tuned | | 0.075 | 0.091 | | |
| | | | from scratch | | 0.773 | - | | |

Table S-1: **Chronicling America Results and Comparisons.** This table reports the performance of different OCR architectures, *off-the-shelf* (without fine-tuning on target data), *fine-tuned* on the Chronicling America training set from initialization on the best public, pre-trained OCR checkpoint, and pre-trained *from scratch* on a consistent, standardized set of synthetic text lines and then fine-tuned on the Chronicling America training set. "?" indicates that the field is unknown due to the proprietary nature of the architecture. Inference speeds are based on an extrapolation from inference speeds measured for EffOCR-Word (Small) to digitize the entire Chronicling America collection using cloud compute.

| | |
|---|---|
| | WASHINGTON, April 1 Ambas- |
| | FORT WORTH JITNEYS OUIT |
| | General Plan  5-4-31 |
| | State of Tennessee |
| | A non-Federal project to furnish free home assistance |
| | SEED DISTRIBUTION |
| | Iron, Steel and Tin Workers |
| | ADVERSE REPORTS ON DEMENTS NOMINATION |
| | IMPROVEMENT IS SHOWN |

Figure S-1: **Examples of textlines in the Day-Per-Decade evaluation set.** Image crops are shown on the left, with their corresponding EffOCR transcriptions (using the final model set used in the American Stories processing pipeline) on the right.

## Legibility

Legibility classification was tested on a set of 100 image crops (50 articles and 50 headlines) sampled randomly from the 1,094-image legibility training set. All legibility images were double-entered. Since the goal was to be cautious in classifying images as illegible, where annotators disagreed the more legible of the two labels was used.

Annotators were instructed to use the following definitions for legibility labeling:

- **Legible**: Greater than 95% of words in an image readable without context from adjacent words.

- **Borderline**: Between 50 and 95% of words in an image readable without context from adjacent words.

- **Illegible**: Less than 50% of words in an image readable without context from adjacent words.

Inter-annotator agreement was 91% between the two annotators. A sample of annotator discrepancies is presented in Figure S-2

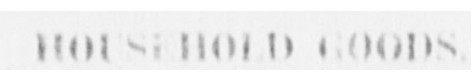

(a) Legible/Borderline                              (b) Borderline/Illegible

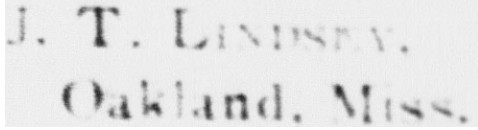

(c) Legible/Borderline                              (d) Borderline/Illegible

Figure S-2: **Examples of Inter-Annotator disagreement in legibility labeling.** Images are labeled "Class 1"/"Class 2" to show the two labeled classes. In cases of disagreement, the more legible of the two labels was used for training and evaluation.

## Applications

The paper presents multiple applications that can be facilitated by the `American Stories` dataset. This section provides details for each application given.

### Topic Classification

To evaluate topic classification, we focused on the topic of politics. As we evaluate at both the scan and article level, for development and test sets we sampled full scans (all articles on the same scan). We took a random sample of up to three front page scans from each election year in our sample. These scans were double-labelled by student research assistants and incongruences were discussed and resolved. We place 20% of these (15 scans, 498 articles) into a development set and the remaining 62 scans (1473 articles) into the test set. Training data was sampled at the article level, rather than the scan level, from the same population of front page articles in election years. Training

data was single-labelled by the same research assistants. A sample were double-labelled to check for consistency and they agreed on the labelling in 93% of cases.

To evaluate neural methods, we finetune RoBERTa large [20] on the training set for ten epochs, with a batch size of 16, and a learning rate of 2e-6.

For evaluation of sparse methods, we use two different methods to select keywords. First, we use the test and evaluation sets to mine keywords. We use TF-IDF to pull words and bigrams that are most commonly found in train set articles about politics, but not found in off-topic articles. We take the top 40 words and bigrams and then sequentially pick those that maximise F1 on the evaluation set, until there is no remaining keyword that increases F1. Using this technique, the mined keywords were: vote, election, republican, committee, united, party, president, congress.

Second, we prompted Chat-GPT to produce keywords. We used the prompt: "You have a large number of 19th century US newspaper articles. You wish to classify these on whether they are about politics or not. The only way that you can do this is by checking whether they match any of a list of keywords or keyphrases. You can search for these keywords or phrases in each article, and if it matches any of them it will be classified as about politics, but if it does not match any, it will not. Please provide a list of keywords and phrases that will correctly classify as many of the articles that are about politics as possible, with a minimal number of off-topic articles classified as on topic" and received the following keywords: President, Congress, Election, Senator, Representative, Governor, Democratic Party, Whig Party, Republican Party, Suffrage, Legislation, Lawmakers, Government, Policy, Bill, Campaign, Debate, Vote, Political Convention, Public Office, Political Reform, Impeachment.

Using these lists of keywords, we consider any article to be predicted as on topic if it contains any of these keywords. We do not take case into account.

The structured data in American Stories allows us to classify at the article level, a significant advantage. However, for comparison with Chronicling America, we also evaluate the same methods at the scan level. A scan is counted as on topic if any article on that page is on topic. For neural methods on Chronicling America, we chunk the text into passages of 256 tokens, as the page OCR is significantly longer than the context window.

**Content Dissemination Networks**

To detect reproduced content, we also compare neural and sparse methods. In this case, the neural methods are only possible with American Stories, whereas sparse methods are possible with both American Stories and Chronicling America.

We evaluate these methods on all front pages from March 1, 1916, a randomly selected day. A single day is chosen because reproduced content tends to be published around the same time, so a single day will have a far higher number of reproduced articles than a random selection of front pages across time. On this day, American Stories contains 114 scans, with 2,354 articles. 1,994 of these were not reproduced, while 360 were reproduced. These 360 comprise 113 distinct articles, with the median reproduced article being reprinted 2 times.

For the neural method, we use the pre-trained neural model from [28]. This is a contrastively-trained bi-encoder finetuned from the MPNet Sentence BERT model [26, 32] on a large, hand-annotated sample of pairs of reproduced historical newspaper articles. [28] find this biencoder is marginally improved by running a cross-encoder over the outputs, but we do not reproduce these results as the cross-encoder is computationally costly for a small gain in performance. They also find that this fine-tuned biencoder model outperforms more generic semantic textual similarity models (eg. [26] by up to 20 percentage points. Thus the model is chosen to maximise accuracy, within a reasonable compute budget. At inference time, article representations are clustered using single linkage clustering to detect reproduced content, and spurious links are pruned using community detection. We use the same distance threshold as in [28].

To enable a comparison to Chronicling America, where the content is only available at a page level, we amalgamate these results by page. A page-pair is counted as positive if they have any article in common. Nonetheless, the page-level evaluation of the neural method requires the data to be split into articles. It cannot be run over the unstructured text in Chronicling America.

Therefore for detecting reproduced content in Chronicling America, where we do not have article texts, we deploy the sparse methods from Viral Texts [30]. Viral Texts was designed specifically for detecting reproduced texts in Chronicling America's noisy page-level OCR by looking for overlapping $n$-gram spans. To compare this method between `American Stories` and Chronicling America, we also run it over the articles in `American Stories` and then amalgamate these results at the page level.

Finally, we deploy the locality-sensitive hashing (LSH) specification from [28] to evaluate the performance of sparse methods on `American Stories`, using the same parameters. In particular we do this because the Viral Texts method is not designed to be run at the article level. As expected, we find that LSH performs better than Viral Texts at the article level, but both methods perform significantly worse than the neural methods, in line with the findings of [28].

**Story Clustering**

Finally, we demonstrate that the content in `American Stories` can be clustered into news stories, following the same story between newspapers and across time. To create clusters of stories, we use a contrastively-trained biencoder.

We train this biencoder using data from `allsides.com`, a modern news website which shows how the same story is written by different newspapers. Articles on `allsides.com` are truncated. Groups of articles on the same story on `allsides.com` were used to create positive pairs. For negatives, we used the fact that each article is labelled with various tags, and also that we know which news source each article came from. For each article we take the article from the same news source, with different topic tags, that had the largest cosine similarity, using the biencoder before finetuning. In the cases where there were no articles with different tags from the same news source, we use articles from a news source which is lifted with the same political leaning. The specification that an article has different tags is important for making sure that articles are actual negatives.

Overall this gave 26,194 unique articles, with 18,382 positive pairs and 18,445 negative pairs. We featurized the data as "headline [sep] article" and we finetuned the biencoder from [28] as in experiments we found that this outperformed finetuning an MPNet Sentence BERT model [26, 32] directly. We optimised hyperparameters using hyperband [17]. The best model was trained fro 9 epochs, on a single GPU, with a batch size of 32, a warm up percent of 0.392. We optimised online contrastive loss [8], with and a loss margin of 0.497.

At inference time, we cluster using single-linkage clustering, with a cosine similarity threshold of 0.92. We control cluster size using leiden community detection [33]. We deduplicate the content using the method outlined in the section above. We take all articles that are reprinted at least five times, and run same story clustering over a year at a time.

## Dataset details

### Dataset URL

The dataset can be found at `https://huggingface.co/datasets/dell-research-harvard/AmericanStories`.

This dataset has structured metadata following `schema.org`, and is readily discoverable.[5]

Training labels for the individual models detailed in this paper are also available, and can be found at `https://huggingface.co/datasets/dell-research-harvard/AmericanStoriesTraining`.

### DOI

The DOI for this dataset is: 10.57967/hf/0757.

---

[5]See `https://search.google.com/test/rich-results/result?id=esZkoGgfOsLlnkrvwx9nSQ` for full metadata.

**License**

The dataset has a Creative Commons CC-BY license.

**Dataset usage**

The dataset is hosted on Hugging Face. Each year in the dataset is divided into a distinct file. The dataset can be easily downloaded using the `datasets` library:

As the dataset is very large, files for specific years can be downloaded by specifying them or users can download all data for all years. Additionally, we provide two options for the output type. The first contains data at the article level, with features like newspaper name, page number, edition, date, headline, byline, and article text. The second contains data at the scan level. It contains information including the scan metadata; all detected content regions like articles, photographs, and adverts; legibility information, and bounding box coordinates.

```python
from datasets import load_dataset

# Download data for the year 1809 at the associated article level (Default)
dataset = load_dataset("dell-research-harvard/AmericanStories",
    "subset_years",
    year_list=["1809", "1810"]
)

# Download and process data for all years at the article level
dataset = load_dataset("dell-research-harvard/AmericanStories",
    "all_years"
)

# Download and process data for 1809 at the scan level
dataset = load_dataset("dell-research-harvard/AmericanStories",
    "subset_years_content_regions",
    year_list=["1809"]
)

# Download ad process data for all years at the scan level
dataset = load_dataset("dell-research-harvard/AmericanStories",
    "all_years_content_regions")
```

Users can find more information on accessing the dataset using the dataset card on Hugging Face.

**Author statement**

We bear all responsibility in case of violation of rights.

**Maintenance Plan**

We have chosen to host the dataset on huggingface as this ensures long-term access and preservation of the dataset.

**Dataset documentation and intended uses**

We follow the datasheets for datasets template [6]. Additionally, we have completed the dataset card on Hugging Face which can be accessed using the link to the dataset on Hugging Face hub [6]

## Reproducibility

Moreover, we have included our responses to The Machine Learning Reproducibility Checklist as outlined in table S-2.

---

[6]https://huggingface.co/datasets/dell-research-harvard/AmericanStories

| **Motivation** |
| :---: |

**For what purpose was the dataset created?** Was there a specific task in mind? Was there a specific gap that needed to be filled? Please provide a description.

The dataset was created to provide researchers with a large, high-quality corpus of structured and transcribed newspaper article texts from historical local American newspapers. These texts provide a massive repository of information about topics ranging from political polarization to the construction of national and cultural identities to the minutiae of the daily lives of people's ancestors. The dataset will be useful to a wide variety of researchers including historians, other social scientists, and NLP practitioners.

**Who created this dataset (e.g., which team, research group) and on behalf of which entity (e.g., company, institution, organization)?**

The dataset was created by a team of researchers at Harvard University, New York University, Northwestern Kellogg School, MIT, and Princeton University, led by Melissa Dell.

**Who funded the creation of the dataset?** If there is an associated grant, please provide the name of the grantor and the grant name and number.

Funding was provided by the Harvard Data Science Initiative, compute credits that Microsoft Azure provided to the Harvard Data Science Initiative, Harvard Catalyst, and the Harvard Economics Department Ken Griffin Fund for Research on Development Economics and Political Economy.

**Any other comments?**

None.

| **Composition** |
| :---: |

**What do the instances that comprise the dataset represent (e.g., documents, photos, people, countries)?** Are there multiple types of instances (e.g., movies, users, and ratings; people and interactions between them; nodes and edges)? Please provide a description.

Dataset instances are detected content regions in newspaper page scans from the Library of Congress's Chronicling America collection. In the cases of article, headline, image caption, and byline regions, a text transcription is included if the page is written in English.

**How many instances are there in total (of each type, if appropriate)?**

Version 0.1.0 of American Stories contains 402 million content regions, 294 million of which include a text transcription.

**Does the dataset contain all possible instances or is it a sample (not necessarily random) of instances from a larger set?** If the dataset is a sample, then what is the larger set? Is the sample representative of the larger set (e.g., geographic coverage)? If so, please describe how this representativeness was validated/verified. If it is not representative of the larger set, please describe why not (e.g., to cover a more diverse range of instances, because instances were withheld or unavailable).

The Version 1.0 of the dataset will contain all possible instances. Version 0.1.0 contains approximately 40% of all instances as of 6/7/23.

**What data does each instance consist of? "Raw" data (e.g., unprocessed text or images) or features?** In either case, please provide a description.

Each instance includes: a unique content region id, its detected class (ARTICLE, HEADLINE, CAPTION, BYLINE, IMAGE, AD, TABLE, HEADER, PAGE NUMBER, or MASTHEAD), and the pixel coordinates of the newspaper page bounding box for the identified region. If the content region is classified as ARTICLE, HEADLINE, CAPTION, or BYLINE, the transcribed text is also provided.

**Is there a label or target associated with each instance?** If so, please provide a description.

Content regions are labeled by their model predicted class. Text article transcriptions have no label.

**Is any information missing from individual instances?** If so, please provide a description, explaining why this information is missing (e.g. because it was unavailable). This does not include intentionally removed information but might include, e.g., redacted text.

No.

**Are relationships between individual instances made explicit (e.g., users' movie ratings, social network links)?** If so, please describe how these relationships are made explicit.

All articles and other content regions include metadata that can definitively determine relationships to other content regions. For example, two articles with the same lccn (newspaper identifier), edition, and page number are from the same newspaper page scan.

**Are there recommended data splits (e.g., training, development/validation, testing)?** If so, please provide a description of these splits, explaining the rationale behind them.

There are no recommended splits.

**Are there any errors, sources of noise, or redundancies in the dataset?** If so, please provide a description.

Layout detection, OCR, and article association all introduce noise.

**Is the dataset self-contained, or does it link to or otherwise rely on external resources (e.g., websites, tweets, other datasets)?** If it links to or relies on external resources, a) are there guarantees that they will exist, and remain constant, over time; b) are there official archival versions of the complete dataset (i.e., including the external resources as they existed at the time the dataset was created); c) are there any restrictions (e.g., licenses, fees) associated with any of the external resources that might apply to a future user? Please provide descriptions of all external resources and any restrictions associated with them, as well as links or other access points, as appropriate.

The provided text data are self-contained. Some applications could require downloading the original scans, which are at `https://chroniclingamerica.loc.gov/`. All scans are freely available and in the public domain.

**Does the dataset contain data that might be considered confidential (e.g., data that is protected by legal privilege or by doctor-patient confidentiality, data that includes the content of individuals non-public communications)?** If so, please provide a description.

The dataset is drawn entirely from image scans in the public domain that are freely available for download from the Library of Congress's website.

**Does the dataset contain data that, if viewed directly, might be offensive, insulting, threatening, or might otherwise cause anxiety?** If so, please describe why.

Texts in the dataset reflect attitudes and values of a large, diverse group of newspaper editors and writers in the period they were written (1790-1960) and include content that may be considered offenseive for a variety of reasons.

**Does the dataset relate to people?** If not, you may skip the remaining questions in this section.

Yes. The dataset contains news about people.

**Does the dataset identify any subpopulations (e.g., by age, gender)?** If so, please describe how these subpopulations are identified and provide a description of their respective distributions within the dataset.

It may be possible to infer certain characters about individuals covered in the news historically from the data. The authors of the dataset do not identify any subpopulations.

**Is it possible to identify individuals (i.e., one or more natural persons), either directly or indirectly (i.e., in combination with other data) from the dataset?** If so, please describe how.

If an individual appeared in the news during this period, then texts may contain their name and other information. In some cases, it may be possible to link individuals to information on ancestry websites or Wikipedia (in the case of prominent historical figures). We do not attempt to do so in this paper.

**Does the dataset contain data that might be considered sensitive in any way (e.g., data that reveals racial or ethnic origins, sexual orientations, religious beliefs, political opinions or union memberships, or locations; financial or health data; biometric or genetic data; forms of government identification, such as social security numbers; criminal history)?** If so, please provide a description.

The data are drawn entirely from newspaper scans in the public domain.

**Any other comments?**

None.

---

**Collection Process**

**How was the data associated with each instance acquired?** Was the data directly observable (e.g., raw text, movie ratings), reported by subjects (e.g., survey responses), or indirectly inferred/derived from other data (e.g., part-of-speech tags, model-based guesses for age or language)? If data was reported by subjects or indirectly inferred/derived from other data, was the data validated/verified? If so, please describe how.

The pipeline used to create layouts and article transcriptions from page images is described in detail within the paper. The dataset described here is the output of that pipeline.

**What mechanisms or procedures were used to collect the data (e.g., hardware apparatus or sensor, manual human curation, software program, software API)?** How were these mechanisms or procedures validated?

The data extraction pipeline is described and evaluated in the main paper text.

**If the dataset is a sample from a larger set, what was the sampling strategy (e.g., deterministic, probabilistic with specific sampling probabilities)?**

Release 1.0 will include everything in the Chronicling America scan collection.

**Who was involved in the data collection process (e.g., students, crowdworkers, contractors) and how were they compensated (e.g., how much were crowdworkers paid)?**

A large group of professors, research assistants, and students collaborated on all aspects of the data collection process, including labeling training data, training and validating models, data engineering, and conceptual design. All were compensated for their work, according to the regulations of Harvard University and New York University.

**Over what timeframe was the data collected? Does this timeframe match the creation timeframe of the data associated with the instances (e.g., recent crawl of old news**

**articles)?** If not, please describe the timeframe in which the data associated with the instances was created.

Scans from Chronicling America were processed between 6/1/23 and 6/7/23. The data associated with the instances were created between 1780 and 1963, when they were published in local newspapers.

**Were any ethical review processes conducted (e.g., by an institutional review board)?** If so, please provide a description of these review processes, including the outcomes, as well as a link or other access point to any supporting documentation.

The data are entirely in the public domain and hence do not fall under the jurisdiction of university institutional review boards.

**Does the dataset relate to people?** If not, you may skip the remaining questions in this section.

Yes, the articles in the dataset talk about people.

**Did you collect the data from the individuals in question directly, or obtain it via third parties or other sources (e.g., websites)?**

We collected this data from a third party, the Library of Congress, which has verified that all data are in the public domain.

**Were the individuals in question notified about the data collection?** If so, please describe (or show with screenshots or other information) how notice was provided, and provide a link or other access point to, or otherwise reproduce, the exact language of the notification itself.

The data are in the public domain and cover many millions of individuals, most of whom are deceased.

**Did the individuals in question consent to the collection and use of their data?** If so, please describe (or show with screenshots or other information) how consent was requested and provided, and provide a link or other access point to, or otherwise reproduce, the exact language to which the individuals consented.

The data are in the public domain and cover many millions of individuals, most of whom are deceased.

**If consent was obtained, were the consenting individuals provided with a mechanism to revoke their consent in the future or for certain uses?** If so, please provide a description, as well as a link or other access point to the mechanism (if appropriate).

Not applicable.

**Has an analysis of the potential impact of the dataset and its use on data subjects (e.g., a data protection impact analysis) been conducted?** If so, please provide a description of this analysis, including the outcomes, as well as a link or other access point to any supporting documentation.

No such analysis has been conducted.

**Any other comments?**

None.

---

| Preprocessing/cleaning/labeling |
|:---:|

**Was any preprocessing/cleaning/labeling of the data done (e.g., discretization or bucketing, tokenization, part-of-speech tagging, SIFT feature extraction, removal of**

**instances, processing of missing values)?** If so, please provide a description. If not, you may skip the remainder of the questions in this section.

No preprocessing was conducted.

---

| Uses |
| :---: |

**Has the dataset been used for any tasks already?** If so, please provide a description.

Example uses are detailed in the main text.

**Is there a repository that links to any or all papers or systems that use the dataset?** If so, please provide a link or other access point.

No such repository currently exists.

**What (other) tasks could the dataset be used for?**

There are a large number of potential uses in the social sciences, digital humanities, and deep learning research, discussed in more detail in the main text.

**Is there anything about the composition of the dataset or the way it was collected and preprocessed/cleaned/labeled that might impact future uses?** For example, is there anything that a future user might need to know to avoid uses that could result in unfair treatment of individuals or groups (e.g., stereotyping, quality of service issues) or other undesirable harms (e.g., financial harms, legal risks) If so, please provide a description. Is there anything a future user could do to mitigate these undesirable harms?

This dataset contains unfiltered content composed by newspaper editors, columnists, and other sources. It reflects their biases and any factual errors that they made.

**Are there tasks for which the dataset should not be used?** If so, please provide a description.

We would urge caution in using the data to train generative language models - without additional filtering - as it contains content that many would consider toxic.

**Any other comments?**

None.

---

| Distribution |
| :---: |

**Will the dataset be distributed to third parties outside of the entity (e.g., company, institution, organization) on behalf of which the dataset was created?** If so, please provide a description.

Yes. The dataset is available for public use.

**How will the dataset will be distributed (e.g., tarball on website, API, GitHub)** Does the dataset have a digital object identifier (DOI)?

The dataset is available via HuggingFace. Download and instructions are at `https://huggingface.co/datasets/dell-research-harvard/AmericanStories`. The dataset's DOI is: https://doi.org/10.57967/hf/0757

**When will the dataset be distributed?**

The dataset is currently available.

**Will the dataset be distributed under a copyright or other intellectual property (IP) license, and/or under applicable terms of use (ToU)?** If so, please describe this license

and/or ToU, and provide a link or other access point to, or otherwise reproduce, any relevant licensing terms or ToU, as well as any fees associated with these restrictions.

The dataset is distributed under a Creative Commons CC-BY license. The terms of this license can be viewed at `https://creativecommons.org/licenses/by/2.0/`

**Have any third parties imposed IP-based or other restrictions on the data associated with the instances?** If so, please describe these restrictions, and provide a link or other access point to, or otherwise reproduce, any relevant licensing terms, as well as any fees associated with these restrictions.

There are no third party IP-based or other restrictions on the data.

**Do any export controls or other regulatory restrictions apply to the dataset or to individual instances?** If so, please describe these restrictions, and provide a link or other access point to, or otherwise reproduce, any supporting documentation.

No export controls or other regulatory restrictions apply to the dataset or to individual instances.

**Any other comments?**

None.

---

**Maintenance**

---

**Who will be supporting/hosting/maintaining the dataset?**

HuggingFace will continue to host the dataset. The authors will provide support, updates, and maintenance.

**How can the owner/curator/manager of the dataset be contacted (e.g., email address)?**

Melissa Dell can be contacted via email at melissadell@fas.harvard.edu

**Is there an erratum?** If so, please provide a link or other access point.

There is no erratum.

**Will the dataset be updated (e.g., to correct labeling errors, add new instances, delete instances)?** If so, please describe how often, by whom, and how updates will be communicated to users (e.g., mailing list, GitHub)?

The dataset will continue to be updated as new scans are processed. New versions will be added to HuggingFace. Anyone can subscribe to notifications about the dataset via HuggingFace.

**If the dataset relates to people, are there applicable limits on the retention of the data associated with the instances (e.g., were individuals in question told that their data would be retained for a fixed period of time and then deleted)?** If so, please describe these limits and explain how they will be enforced.

All data are in the public domain.

**Will older versions of the dataset continue to be supported/hosted/maintained?** If so, please describe how. If not, please describe how its obsolescence will be communicated to users.

Older versions of the dataset will still be visable via the HuggingFace repo.

**If others want to extend/augment/build on/contribute to the dataset, is there a mechanism for them to do so?** If so, please provide a description. Will these contributions be validated/verified? If so, please describe how. If not, why not? Is there a process

for communicating/distributing these contributions to other users? If so, please provide a description.

The dataset is privately created and maintained. There is no current plan to allow open source contributions.

**Any other comments?**

None.

Table S-2: Reproducibility checklist

| Item | Response | Comment |
| --- | --- | --- |
| **For all models and algorithms presented, check if you include:** | | |
| A clear description of the mathematical setting, algorithm, and/or model. | Yes | |
| A clear explanation of any assumptions. | Yes | |
| An analysis of the complexity (time, space, sample size) of any algorithm. | NA | Complexity can depend upon application-specific architecture and methods. We are using transformer-based models within our framework and we have reported the time profile and other related details |
| **For any theoretical claim, check if you include:** | | |
| A clear statement of the claim. | Yes | |
| A complete proof of the claim. | Yes | |
| **For all datasets used, check if you include:** | | |
| The relevant statistics, such as number of examples | Yes | |
| The details of train / validation / test splits | Yes | |
| An explanation of any data that were excluded, and all pre-processing step. | Yes | |
| A link to a downloadable version of the dataset or simulation environment | Yes | Link to Hugging Face Hub repo provided |
| For new data collected, a complete description of the data collection process, such as instructions to annotators and methods for quality control. | Yes | |
| **For all shared code related to this work, check if you include:** | | |
| Specification of dependencies. | Yes | |
| Training code. | Yes | |
| Evaluation codes | Yes | |
| (Pre-)trained model(s). | Yes | Available on HuggingFace |
| README file includes table of results accompanied by precise command to run to produce those results | Yes | |
| **For all reported experimental results, check if you include:** | | |
| The range of hyper-parameters considered, method to select the best hyper-parameter configuration, and specification of all hyper-parameters used to generate results | Yes | |
| The exact number of training and evaluation runs. | Yes | |
| A clear definition of the specific measure or statistics used to report results. | Yes | |

| | | |
|---|---|---|
| A description of results with central tendency (e.g. mean) variation (e.g. error bars). | NA | |
| The average runtime for each result, or estimated energy cost. | Yes | Both training and inference times have been reported |
| A description of the computing infrastructure used | Yes | |

