| Image | Transcription |
|---|---|
| WASHINGTON, April 1—Ambas- | WASHINGTON, April 1 Ambas- |
| FORT WORTH JITNEYS QUIT | FORT WORTH JITNEYS OUIT |
| General Plan 5- 4-31 | General Plan 5-4-31 |
| State of Tennessee, | State of Tennessee |
| A non-Federal project to furnish free home assistance | A non-Federal project to furnish free home assistance |
| SEED DISTRIBUTION | SEED DISTRIBUTION |
| Iron. Steel and Tin Workers | Iron, Steel and Tin Workers |
| ADVERSE REPORTS ON DEMENT'S NOMINATION. | ADVERSE REPORTS ON DEMENTS NOMINATION |
| IMPROVEMENT IS SHOWN | IMPROVEMENT IS SHOWN |

Figure 1: **Examples of textlines in the Day-Per-Decade evaluation set.** Image crops are shown on the left, with their corresponding EffOCR transcriptions (using the final model set used in the American Stories processing pipeline) on the right.

| Model/Engine | Seq2Seq? | Transformer? | Pretraining | Parameters | CER [2] | CER Day-Per-Decade | Lines Per Second | Cost Per 10K Lines |
|---|---|---|---|---|---|---|---|---|
| EffOCR-C (Base) | × | × | from scratch | 112.5 M | 0.023 | 0.062 | 0.27 | $1.77 |
| EffOCR-C (Small) | × | × | from scratch | 9.3 M | 0.028 | 0.080 | 7.28 | $0.06 |
| EffOCR-T (Base) | × | ✓ | from scratch | 101.8 M | 0.022 | 0.059 | 0.17 | $2.80 |
| **EffOCR-Word (Small)** | × | × | **from scratch** | **10.6 M** | **0.015** | **0.043** | **11.60** | **$0.04** |
| Google Cloud Vision OCR | ? | ? | off-the-shelf | ? | 0.005 | 0.019 | ? | $15.00 |
| Tesseract OCR (Best) | ✓ | × | off-the-shelf | 1.4 M | 0.106 | 0.170 | 2.43 | $0.19 |
| EasyOCR CRNN | ✓ | × | off-the-shelf fine-tuned from scratch | 3.8 M | 0.170 0.036 0.131 | 0.274 0.157 - | 10.75 | $0.04 |
| PaddleOCR SVTR | × | × | off-the-shelf fine-tuned from scratch | 11 M | 0.304 0.103 0.104 | | 7.36 | $0.06 |
| TrOCR (Base) | ✓ | ✓ | off-the-shelf fine-tuned from scratch | 334 M | 0.015 0.013 0.809 | 0.038 0.027 - | 0.23 | $2.02 |
| TrOCR (Small) | ✓ | ✓ | off-the-shelf fine-tuned from scratch | 62 M | 0.039 0.075 0.773 | 0.121 0.091 - | 0.53 | $0.90 |