# OpenReview forum: "American Stories: A Large-Scale Structured Text Dataset of Historical U.S. Newspapers"
_NeurIPS.cc/2023/Track/Datasets_and_Benchmarks — NeurIPS 2023 Datasets and Benchmarks Poster_

### Official Review · Reviewer_1942 · 2023-07-02
**American Stories: A Large-Scale Structured Text Dataset of Historical U.S. Newspapers**

**Rating:** 6
**Confidence:** 4

**Strengths:**

- The paper introduces a deep learning-based pipeline for extracting full article texts from American newspaper images.
- This pipeline addresses multiple tasks , such as layout detection, legibility classification, and OCRing which of course offers a holistic approach to document image analysis.
- The authors introduce a dataset named American Stories which is useful in training and evaluating multimodal document layout analysis models

**Additional Feedback:**

-

**Clarity:**

The paper lacks clarity and coherence, making it difficult to follow. Some ideas are too generic, and the supplemental material does not provide clear information on the specific machine learning tasks for which the dataset is explicitly used.

**Correctness:**

The paper provides a comprehensive discussion on the data collection and preparation process. The DIA pipeline presented in the paper is described, although there is no clear differentiation from the work mentioned in reference [12].

**Documentation:**

Overall, the data-set collection and organization are well described. However, some information is overlooked, such as the intended tasks and specific instance details of the dataset, which are not clearly stated. A link to access the dataset is provided as well.

**Ethics:**

Since the dataset is composed from Historical newspaper there might raise an ethical concerns such as personal information, and cultural sensitivity.

**Limitations:**

The author has not provided the limitations of their work. Given the involvement of multiple individuals and historical document information, it is crucial to acknowledge and address potential societal negative impact that may arise.

**Opportunities For Improvement:**

- The exact contribution of this paper in comparison to the 2020 publication titled "The Newspaper Navigator Dataset: Extracting Headlines and Visual Content from 16 Million Historic Newspaper Pages in Chronicling America" (ref: Contribution section of [12]) is not clearly stated.
- It would be beneficial for the author to include the checklist provided in the NeurIPS conference template. Additionally, the answers given in the datasheet are often unclear and lack conciseness.
- The real advantages of legibility classification are not clearly explained in the paper.
- It would be helpful if the author explicitly listed the specific tasks for which this dataset could be used in machine learning applications, even if those tasks are not provided in the supplementary material or the main paper.
- The paper covers multiple tasks but fails to mention the number of instances and corresponding ground-truth information, such as bounding boxes or text transcriptions.
- The notability of the presented pipeline in the paper is questionable, as there are already numerous open-source Document Image Analysis (DIA) tools available e.g the one given in conference [12].

**Relation To Prior Work:**

The author attempts to discuss how the current work differs from the previous one (https://arxiv.org/pdf/2005.01583.pdf), but I am not convinced of the contribution of their work, as it appears to be almost identical to the prior work.

**Summary And Contributions:**

This paper proposes a new deep learning-based pipeline for extracting full article texts from American newspaper images. The pipeline is useful for several tasks, including layout detection, legibility classification, and OCRing. In addtion, the authors also introduce a new dataset named American Stories that  can be valuable for multimodal layout analysis models and other multimodal applications in document image analysis.

---

> ### Author Response · Authors · 2023-08-22
> **Author response**
>
> We thank the reviewer for their thoughtful comments. We direct all reviewers to our updated submission, and respond individually here to important points raised in this review.
>
> *The reviewer raises several concerns regarding our contribution relative to "The Newspaper Navigator Dataset" (ref: [12])*
>
> Response: The main, and crucial, difference between American Stories and [12] is that [12]  does not detect bounding boxes of articles. [12] focuses on ``7 classes of visual content: headlines, photographs, illustrations, maps, comics, editorial cartoons, and advertisements’’. Distinguishing articles enables legibility classification, application of custom OCR, and association of articles across bounding boxes. These tasks are at the core of our contribution, and enable the usefulness of our contribution for downstream applications (Section 6). In addition, the OCR in [12] is limited to ChroniclingAmerica’s OCR. We show in Figure 6 that this leads to significant quality deterioration. We have now updated the  text to make these distinctions more salient.
>
> *Supplementary materials*
>
> Response: We now include the NeurIPS checklist in the supplementary materials and have edited the datasheet for clarity and concision. We have added details about the applications to the Supplementary Materials, so that they can be reproduced.
>
> *Downstream machine learning tasks: It would be helpful if the author explicitly listed the specific tasks for which this dataset could be used in machine learning applications*
>
> Response: There are several canonical machine learning applications that American Stories can be used for. These include:
> - The document layout annotations assembled can be used to train or assess models for layout analysis and document understanding.
> - The high quality, structured text transcriptions can be used to train or fine-tune language models.
> - Image-caption pairs could be used for training and assessing image captioning models, visual question-answering models, cross-modal retrieval models, image retrieval models, and multi-modal understanding (e.g., representation learning) models.
>
> The American Stories dataset can also be leveraged to substantially lower the costs of creating new datasets and benchmarks for other common ML tasks, e.g.,image and text classification, named entity recognition, and image retrieval
>
> *Editing, exposition, etc.: The reviewer asks for more clarity of exposition in the main text, and the supplementary.*
>
> Response: The paper has been edited for clarity. The datasheet has been edited for conciseness.
>
> *The paper covers multiple tasks but fails to mention the number of instances and corresponding ground-truth information, such as bounding boxes or text transcriptions.*
>
> Response: These numbers are now provided in Section 5.
>
> *Legibility classification: The real advantages of legibility classification are not clearly explained in the paper.*
>
> Response: Substantial shares of illegible content can degrade language model training and influence downstream applications. For example, it is common for social scientists to construct data based on the presence of keyword terms, assigning a positive outcome if the term is present and a zero outcome otherwise. Illegibility can bias downstream analyses if it is correlated with an underlying unobserved factor. See [7] for examples of papers that use content counts without accounting for legibility. We now include Figures 3(a) and 3(b) showing that the share of illegible content varies substantially across both space and time. Hence, it is likely to be correlated with various unobserved variables that could bias downstream analyses, which also vary across space and time.
>
> *Broader impact: The reviewer raises concerns around societal impact, especially around individual information and the use of historical documents.*
>
> Response: These issues are very important. All newspaper scans we use are in the public domain. In addition, as we now explicitly discuss in the paper, the United States government uses the 72 year rule for release of personal information into the public domain, most notably from the decadal censuses, which include income and a host of other individual and household characteristics. The vast majority of our newspaper scans are older than that, with less than 2% of content produced after 1951.

---

### Official Review · Reviewer_Z8NQ · 2023-07-21
**Review of American Stories**

**Rating:** 5
**Confidence:** 3

**Strengths:**

* large-scale dataset with various possible applications
* efforts to minimise the noise from automation

**Additional Feedback:**

L106: If illegibility is not random, this may be important. What is the variability across space and time?

L137: Why only from these centuries?

L145: Not clear to me, please explain what non-max suppression is.

L147: Do you have experimental results to show this major advantage?

L151: Please provide an example figure.

L153: How was the model selection done, e.g., why not Yolo v8?

L165-166: How does this augmentation work exactly?

L170-180: Why did you remove 3,999 words (how is it a problem if they remained)?

L170-180: How did you compute the (34%) savings in the deployment costs?

L184 (word crop): Is this defined somewhere? I assume that this is a sequence of extracted characters, but is it whitespace-bounded or otherwise?

L185 (0.82): How is this defined? How many mistakes are there in a random sample of 1,000 words (above 0.82) or so?

L185 (embeddings): How are words embedded exactly? Using TFIDF, dense representations, otherwise?

L185-186 (we default to…): Not entirely clear to me. So, you detect boxes comprising words, you apply image retrieval, and, when the similarity is low, you apply character-level OCR on the detected image?

L188 (synthetic word renders): Not clear to me. Is this the same word, depicted with 43 different fonts? Which fonts? And do the newspapers use that many different fonts to motivate this idea?

L191 (is accurate): Where is this shown in the paper?

L195: Where is this shown?

L204: Please share more details.

L206 (randomly selected): From which years, of what type, etc. Please show statistics.

L207: How many discrepancies and what was their cause? What is the inter-annotator agreement? What is the annotators’ opinion for the quality of the texts?

L209 (non-word rate): Not defined yet.

L225: Please define confusability and show results for the borderline class.

L231: Given that the script exists, please apply it and report the drop in CER.

L235: What are the non-words which are present? And what is the vertical axis? If it is pages, the non-word rate is not low.

L275: Dense description of methods, with model selection being vague (are these baselines or the state of the art).

L287-291: Where is this result? I don’t see Table 5 in the main text.

**Clarity:**

* Many missing details (please see my feedback).
* The description of the methods (in the applications) is dense and belongs to a section discussing potential applications with no experimental results (could be much shorter to give space for details on your experiments).

**Correctness:**

Methods are claimed accurate (EfficientOCR in L191, rule-based content association in L195) but not benchmarked. On the contrary, only the selected methods (of the presented pipeline) are assessed. Also, an inter-annotator agreement study should complement the dataset development.

**Documentation:**

The training dataset for the custom OCR was vaguely described (L188-191), making it non-reproducible.

**Limitations:**

* The quality of the extracted text is not verified.
* Inter-annotator agreement not discussed.
* Limited benchmark: for each task, I would expect to see some comparison between methods (e.g., custom OCR vs EfficientOCR).

**Opportunities For Improvement:**

The presentation could be improved:
* Many missing details (please see my feedback).
* The methods (in the applications) have a dense description while I couldn't find any Table 5 in the paper (typo for Table 3?).

**Relation To Prior Work:**

There wasn’t any section discussing related work and the differences from previous work are not clear.

**Summary And Contributions:**

The paper presents a structured dataset of OCRed text from approx. 20m scanned historical US newspapers. The dataset is created by a custom digitization pipeline, performing independently layout/line detection, cleaning, OCR, and article linking. A yolo v8 classifier was trained (fine-tuned?) on 48,874 layout objects from 2,202 images in newspapers of the 19th and 20th cent, to yield parts of the image that are Tables, Articles, Headers, etc. A mobilenet classifier was trained (or fine-tuned, not clear to me) on 1,094 labelled content regions (Article, Headline, Caption Byline) to classify each as legible, borderline, illegible. EfficientOCR was used for OCR, combining character- and word-level retrieval to speed-up the process. A word lexicon was formed and used. For evaluation purposes, ten pages were manually annotated and the performance per task was measured. The paper also comprises an applications section, showing potential applications and experimenting with one: that of detecting reproduced content, which was approached with clustering.

---

> ### Author Response · Authors · 2023-08-22
> **Author Response 1**
>
> We thank reviewer Z8NQ for their thoughtful questions, which raised a number of important points. We direct all reviewers to our updated submission, and respond individually here to important points raised in this review.
>
> **Broader Points:**
> *The quality of the extracted text is not verified.*
>
> Response: Table 2 and Supplementary Table 1 summarize our validation exercises. We manually annotate and compute the end-to-end character error rate for 10 full page scans (Table 2), and additionally validate our selected OCR architecture over a day-per-decade textline sample and a separate OCR validation sample (Supplementary Table 1). Before post processing, we observe a 5.1% end-to-end pipeline CER, which drops to 4.4% after post processing and spell checking.
>
> *Inter-annotator agreement not discussed.*
>
> Response: In the context of legibility classification, inter-annotator agreement was 89% (n = 671 images). All but one of the disagreements were Legible/Questionable or Questionable/Illegible. Since our goal was to provide as much content as reasonably possible, we used the more legible label of the two when training our legibility classifier. A selection of annotator discrepancies is available as Supplementary Figure 1 in the supplementary materials. The bounding box and OCR labels were also completed by two highly skilled undergraduate research assistants and all discrepancies were resolved by hand. In these cases, the ground truth is easily verifiable and not a place where there is scope for any disagreement. These annotations are publicly available at https://huggingface.co/datasets/dell-research-harvard/AmericanStoriesTraining.
>
> *“Limited benchmark: for each task, I would expect to see some comparison between methods (e.g., custom OCR vs EfficientOCR).”*
>
> Response: Supplementary Table 1 in the supplemental materials now compares the character error rate, speed, and cost estimates of a wide variety of open-source and proprietary OCR engines on Chronicling America, reproducing this table from Carlson et al. (2023). We show that EffficientOCR is the most accurate OCR engine that meets our cost constraints. TrOCR Base, a large open-source OCR model, is modestly more accurate, but would cost 50 times more to deploy. Google Cloud Vision (GCV) is significantly more accurate. However, GCV makes significant layout errors when fed full newspaper page scans, which have complex layouts (Shen et al., 2021), and hence the line-level performance in Supplementary Table 1 cannot be achieved when it is fed scans. GCV charges per image, and the supplementary materials estimate a cost at current prices of 23 million USD to digitize LoCCA at the line image level, over several orders of magnitude more costly than our 60K budget.
>
> *The description of the methods (in the applications) is dense and belongs to a section discussing potential applications with no experimental results*
>
> Response: We have added detail on these methods in a separate appendix, to allow for clearer explanations, with the key points in the main paper.
>
> *The training dataset for the custom OCR was vaguely described (L188-191), making it non-reproducible.*
>
> Response: This dataset is now available at: https://huggingface.co/datasets/dell-research-harvard/AmericanStoriesTraining, with a detailed description provided in the supplementary materials.
>
> *There wasn’t any section discussing related work and the differences from previous work are not clear.*
>
> Response: We now include a section explicitly on related work. The most closely related work is "The Newspaper Navigator Dataset: Extracting Headlines and Visual Content from 16 Million Historic Newspaper Pages in Chronicling America" (ref: [12]) The main, and crucial, difference between American Stories and [12] is that [12]  does not detect bounding boxes of (parts of) articles. [12] focuses on ``7 classes of visual content: headlines, photographs, illustrations, maps, comics, editorial cartoons, and advertisements’’. Distinguishing articles enables legibility classification, application of custom OCR, and association of articles across bounding boxes. These tasks are at the core of our contribution, and enable the usefulness of our contribution for downstream applications. In addition, because [12] does not detect articles, their OCR is limited to ChroniclingAmerica’s OCR. We show in Figure 3 that this leads to significant quality deterioration.

---

> ### Author Response · Authors · 2023-08-22
> **Author Response 2**
>
> **Line Items:**
>
> *L106: If illegibility is not random, this may be important. What is the variability across space and time?*
>
> Legibility does indeed vary across space and time, making it an important consideration for content-based analyses. We have added figures 3(a) and 3(b) to the supplementary materials to visualize this variation.
>
> *L137: Why only from these centuries?*
>
> We sought to annotate newspapers representative of the Chronicling America corpus. These are the centuries almost exclusively represented in the corpus, with less than 0.1% is from the 18th century.
>
> *L145: Not clear to me, please explain what non-max suppression is.*
>
> Non-max suppression chooses a single entity out of many overlapping entities and is an important hyperparameter in object detection methods such as Yolo v8 or Cascade RCNN.
>
> *L147: Do you have experimental results to show this major advantage?*
>
> This is based on our own extensive experience working with Tesseract OCR (which uses rule-based line detection) on over 12 million newspaper scans from a later period. When the bounding boxes around articles have even minimal noise, it sends line detection haywire, leading the last line of an article to be frequently OCR’ed as “eeee” or other garbage characters. However, because we don’t have space to properly explain and evaluate this claim, we have removed it.
>
> *L151: Please provide an example figure.*
>
> One example of a newspaper page from the collection with very small fonts can be found here: https://chroniclingamerica.loc.gov/data/batches/mnhi_plymouth_ver01/data/sn90059523/00206537826/1904062001/0298.pdf
>
> *L153: How was the model selection done, e.g., why not Yolo v8?*
>
> We trained a variety of image classification models on the legibility classification task, including YOLO, convnextTiny, and mobileVIT. MobileNetv3 (Small) met our criteria of producing no Legible/Illegible errors on the validation set, as well as >90% overall validation accuracy. In addition, we measured it inferencing at >7x faster than Yolov8.
>
> *L165-166: How does this augmentation work exactly?*
>
> We deploy several image augmentations and the augmentation scheme slightly differed for words and characters, using transformations provided in the Torchvision library. These include Affine transformation (only slight translation and scaling allowed), Random Color Jitter, Random Autocontrast, Random Gaussian Blurring, and Random Grayscale. Additionally, we pad the character to make the image square while preserving the aspect ratio of the character render.
> We do not use common augmentations like Random Cropping or Center Cropping, to avoid destroying too much information. We slightly modify the data augmentation scheme for words to account for a different aspect ratio and variation in word crops. We allow for a slight (-5 to +5 degree) rotation to account for slight skewness  imperfect localisation and also add more transformations tailored to this use case - Random Equalize, Random Posterize, Random Solarize, Random Inversion and Random Erase (randomly erase 0-5\% of the image).
>
> *L170-180: Why did you remove 3,999 words (how is it a problem if they remained)?*
>
> We removed words we observed to never appear in a corpus of historical American English. Manual validation indicated that these were overwhelming words exclusive to modern contexts, e.g., terminology from computers or modern medicine. Each word in the training vocabulary which does not appear in the underlying data increases the chance of a given word being misclassified.
>
> *L170-180: How did you compute the (34%) savings in the deployment costs?*
>
> We measured the relative speeds of EfficientOCR with only character recognition vs. with our word recognition strategy on a sample dataset, and found that the word-level version increased inference speed by 39%. Since OCR constituted 89% of the overall processing time, this results in a 34% overall time saving. Our processing costs are per time-unit of compute used, making time savings directly correspond to cost savings.
>
> *L184 (word crop): Is this defined somewhere? I assume that this is a sequence of extracted characters, but is it whitespace-bounded or otherwise?*
>
> Word crops in this context are words detected by an object detection model trained to localize printed words in an image, in this case YoloV8. Our labels, which are downloadable at https://huggingface.co/datasets/dell-research-harvard/AmericanStoriesTraining, create as close a crop as possible excluding whitespace bounding.

---

> ### Author Response · Authors · 2023-08-22
> **Author Response 3**
>
> *L185 (embeddings): How are words embedded exactly? Using TFIDF, dense representations, otherwise?*
>
> Rather than their use in language modeling contexts, “words embeddings” refer to dense embeddings of image crops of words produced by an image encoder (in this case our contrastively-trained mobileNetv3 word recognition model, described in detail in the supplemental materials). We have clarified the language in the paper to better describe this distinction.
>
> *L185-186 (we default to…): Not entirely clear to me. So, you detect boxes comprising words, you apply image retrieval, and, when the similarity is low, you apply character-level OCR on the detected image?*
>
> Yes, this is exactly correct. We have clarified this explanation.
>
> *L188 (synthetic word renders): Not clear to me. Is this the same word, depicted with 43 different fonts? Which fonts? And do the newspapers use that many different fonts to motivate this idea?*
>
> While image level augmentations from torchvisions can learn a lot about variations like scan quality, noise and background - they can’t handle variation in typography - the skeleton of each character/word. With a period spanning over multiple decades, along with the evolution of printing technology, design preferences also changed. Moreover, different parts of an article tend to have different kinds of fonts (For example, Serif fonts for the headlings and Sans Serif for the article body or vice-versa). Newspapers also tend to use non-standard display fonts at times to capture the reader’s  attention (For example, consider the headline “BREAKING NEWS”). We used 36 fonts for our augmentations.Since our OCR recognizer’s architecture required us to map word/character crops from newspapers to a “clean” render of the character or word, we used NotoSerif-Regular. In order to account for differences in printing technologies, we got free fonts from the internet that mimicked the printing in old newspapers; e.g., a font called OldNewsPapertypes. Headlines also tended to have systematically different fonts than the body (beyond just the serif and sans-serif dichotomy). Often, newspapers used tall, thin “Condensed” fonts. In order to account for all of these style differences, we used a wide-variety of 36 fonts : ['AmcapEternal', 'Arabella', 'BluefishBlackScratchedDemo', 'BluefishScratchedDemo', 'CreamySugar', 'DamageplanPersonalUseBold', 'DeatheMaachNcv', 'Fatcow', 'FatcowItalic', 'UbuntuMono', 'VollkornBlack', 'VollkornBlackItalic', 'VollkornBold', 'VollkornBoldItalic', 'VollkornExtrabold', 'VollkornExtraboldItalic', 'VollkornItalic', 'VollkornMedium', 'VollkornMediumItalic', 'VollkornRegular', 'VollkornSemibold', 'VollkornSemiboldItalic', 'EBGaramond', 'IMFellDWPica', 'Oldnewspapertypes', 'Fredoka', 'NotoSerif', 'Orbitron', 'Ultra', 'VT323', 'NewYorker', 'Anton', 'ZaiCourierPolski1941', 'SpecialElite', 'CutiveMono', 'ZaiConsulPolishTypewriter']. These fonts are all open-source and we have included them in our repo.
>
> *L191 (is accurate): Where is this shown in the paper?*
>
> OCR accuracy results for various architectures, including EffOCR-character, have been added in the supplemental materials as Supplementary Table 1. EffOCR-character achieved a 2.6XX% Character Error Rate in testing on historical newspapers.
>
> *L195: Where is this shown?*
>
> We have clarified the text to make this point clearer. This follows from the fact that only 3.8% of articles span multiple article bounding boxes. Therefore we have a full article accuracy rate of 96.2% without associating article bounding boxes. The full evaluation on the labeled sample, including the association of headlines and bylines, is given in section 5.
>
> *L204: Please share more details.*
>
> This is a model that combines layout information in a rule-based manner, with language understanding from a finetuned RoBERTa base cross-encoder. We construct data to train the cross-encoder based on article bounding boxes that are below the same headline, and directly above the bottom of the page, as these are sequential bounding boxes from the same article with extremely high probability. We prune the number of pairs to compare within a page using rule-based methods (article X does not follow article Y if it is above or to the left of Y) and compare the remaining pairs using the cross-encoder. Full details can be found in Silcock and Dell, 2023.
>
> *L206 (randomly selected): From which years, of what type, etc. Please show statistics.*
>
> The ten labeled scans were selected from across the Chronicling America collection. The ten scans are now posted to the training data collection (https://huggingface.co/datasets/dell-research-harvard/AmericanStoriesTraining/upload/main) under “Gold Data,” and may be downloaded and viewed. All gold scans were published between 1880 and 1920, coinciding with the majority of the Chronicling America collection.

---

> ### Author Response · Authors · 2023-08-22
> **Author Response 4**
>
> *L209 (non-word rate): Not defined yet.*
>
> This has been corrected.
>
> *L225: Please define confusability and show results for the borderline class.*
>
> This refers to legible scans misclassified as illegible, or vice versa. Supplementary Figure 2 provides examples of disagreement over the borderline class.
>
> *L231: Given that the script exists, please apply it and report the drop in CER.*
>
> We have added this result to Table 2 in the text. When applying a spell checker, CER drops from 0.051 to 0.044 overall. CER on headlines increases, reflecting the larger proportion of proper nouns (which may be erroneously “corrected” to other words) in newspaper headlines.
>
> *L235: What are the non-words which are present? And what is the vertical axis? If it is pages, the non-word rate is not low.*
>
> There are many instances when a non-word can be present with a correct transcription; e.g., proper nouns, hyphenated words at the end of lines, and antiquated words that do not show up in our modern word dictionary.
>
> The vertical axis is the number of scans within the Day-per-Decade (n=3,212 total scans) falling into each word rate bucket.
>
> *L275: Dense description of methods, with model selection being vague (are these baselines or the state of the art).*
>
> The selected neural biencoder model is one that has specifically been finetuned for this purpose. Silcock et al. find that this significantly outperforms sparse methods (n-gram overlap, hashing). We confirm these results on American Stories in table 3. Silcock et al. find the biencoder is marginally outperformed by a biencoder + cross-encoder method, but we do not reproduce these results as the cross-encoder is computationally costly for a small gain in performance. They also find that this fine-tuned biencoder significantly outperforms generic semantic textual similarity models on this task, such as S-BERT STS (Riemers et al, 2019). On American Stories, we further find that the biencoder method substantially outperforms other sparse methods that have been designed specifically for this task (such as the Viral Texts project, Smith et al.), also shown in table 3. We have added a section in the appendix with additional details.
>
> *L287-291: Where is this result? I don’t see Table 5 in the main text.*
> This typo has been corrected, and now reads Table 3.

---

### Official Review · Reviewer_CqsX · 2023-07-22
**A new large-scale dataset of structured text and layout of historical newspaper articles**

**Rating:** 6
**Confidence:** 4
**Clarity:** The paper is well written.

**Strengths:**

(1) The proposed American Stories provides a massive corpus for the community, which containing 20 million scans, around 400 million bounding boxes of content regions and 120 million articles.
(2) The texts cover 19th and early 20th century English, providing linguistic diversity.
(3) The annotation of this dataset is comprehensive, encompassing both layout and OCR data.
(4) The paired images, texts, and layouts support multimodal research in areas like layout analysis.

**Additional Feedback:**

It is better to add a section to introduce and review related works.

**Correctness:**

The dataset is constructed in a sound way, but the quality of OCR needs to be guaranteed.

**Documentation:**

This dataset provides details on data collection and organization, usability and maintenance, and ethical and responsible use.

**Ethics:**

No, there is no ethics issue.

**Limitations:**

- This paper focuses solely on news articles, does not cover other document types.
- Most content is from early 20th century, lacks modern texts.
- No evaluation on downstream tasks is given. It is unclear how useful the proposed dataset for various downstream tasks such as language modeling or multimodal layout analysis.
- Models tested are competent but not state-of-the-art.

**Opportunities For Improvement:**

(1) The evaluation of the pipeline was conducted based on a limited dataset comprising only ten manually annotated scans, which is insufficient to precisely represent its performance.
(2)  The paper acknowledges the potential of post-processing in enhancing OCR quality; however, it fails to provide any metrics to demonstrate the impact of post-processing on the final output.
(3) The authors mentioned Table 4 in line 254, but I cannot find Table anywhere. Where is it?
(4) The authors mentioned Table 5 in line 289, where is it?

**Relation To Prior Work:**

This paper does not present a comprehensive introduction to prior works, and does not discuss how this work differs from previous contributions. It merely provides a concise description of a previous similar work.

**Summary And Contributions:**

This paper introduces American Stories, a new large-scale dataset of structured historical newspaper article texts and layouts extracted from the Chronicling America collection. The authors have developed a modular deep learning pipeline to extract article texts and layout regions from newspaper page images. The pipeline involves layout detection, legibility classification, custom OCR, and article text association. As a result, a large-scale structured text dataset of historical U.S. Newspapers, named American Stories, is generated.

---

> ### Author Response · Authors · 2023-08-22
> **Author Response**
>
> We thank the reviewer for their thoughtful comments. We direct all reviewers to our updated submission, and respond individually here to important points raised in this review.
>
> *The evaluation of the pipeline was conducted based on a limited dataset comprising only ten manually annotated scans...the quality of OCR needs to be guaranteed*
>
> Response: To evaluate the accuracy of the end-to-end pipeline, we needed to transcribe all content on the page, since the document layout analysis is applied at the page level. Given an average of nearly 20K characters per scan, double entering all of this (oftentimes difficult to read) text was very costly. We agree that a more diverse evaluation sample is important and hence have two other samples on which we now evaluate the OCR:
>
> - A day per decade sample, where we label 50 randomly selected lines of text per decade. We start with the 1850s, as the overwhelming share of content is post-1850 (less than 0.1% of content comes from the 18th century). These evaluations are presented in Supplementary Table 1, and we also give example line images from the sample decade in Supplementary Figure 1. OCR accuracy improves significantly over time, as the older scans are much more likely to have been poorly preserved or to have unusual visual signatures. Note that the majority of the content comes from the later decades (Figure 1), where OCR quality is higher.
> - A randomly selected sample of 64 lines (1727 characters) from the period 1890-1920, that is taken from Carlson et al., (2023) and used to compare our pipeline to other open-source and proprietary OCR engines. These evaluations are likewise in Supplementary Table 1.
>
> The results in Supplementary Table 1 document that EffOCR is the most accurate engine given our budget constraints. TrOCR Base, a large open-source OCR model, is modestly more accurate, but would cost 50 times more to deploy. Google Cloud Vision (GCV) is significantly more accurate. However, GCV makes significant layout errors when fed full newspaper page scans, which have complex layouts (Shen et al., 2021), and hence the line-level performance in Supplementary Table 1 cannot be achieved when it is fed scans. GCV charges per image, and the supplementary materials estimate a cost at current prices of 23 million USD to digitize LoCCA at the line image level, over several orders of magnitude more costly than our 60K USD budget.
>
> *Post-processing metrics*
>
> Response: We now include an evaluation on post-processed text in Table 2, where transcribed texts are checked using the lightweight SymSpell spellchecker. We see an overall improvement in text quality, although the character error rate does increase slightly in headlines. This appears to be due to the higher concentration of proper nouns, which may be erroneously “corrected” to other words.
>
> *Table number reference errors*
>
> Response: We apologize for creating confusion and have fixed these errors.
>
> *This paper focuses solely on news articles, does not cover other document types. Most content is from early 20th century, lacks modern texts.*
>
> Response: There are already many modern text datasets, and we see our historical data as complementing the overwhelming focus of the literature on modern web texts. Newspapers historically served a purpose in many ways similar to the internet today, containing a vast diversity of content that ranged from international affairs to who had dinner with whom in the town recently. Newspapers are of considerable interest to the public and have generated a large academic literature ([6] provides a review), making them of central interest to many end users. We hope that the success of American Stories will help make research funding available to digitize more public domain archival collections.
>
> *No evaluation on downstream tasks is given.*
>
> Response: We have expanded the downstream evaluation section (Section 6), which now considers three common downstream tasks: topic classification, detection of reproduced content, and story clustering. We show that American Stories supports tasks that are impossible with the existing Chronicling America page level dataset, and significantly improves performance on tasks for which the Chronicling America dataset has been widely used. We also discuss additional applications that we cannot expand upon due to space constraints.
>
> *Models tested are competent but not state-of-the-art.*
>
> Response: The newly-added Supplementary Table 1 in the supplementary materials presents speed and accuracy results for a variety of OCR architectures fine-tuned on historical newspapers, of which EfficientOCR performs best of the computationally feasible options. TrOCR, the state-of-the-art open-source architecture, is around 50 times more costly to deploy.
>
> *It is better to add a section to introduce and review related works.*
>
> Response: We now include a section explicitly discussing related works.

---

> > ### Comment · Reviewer_CqsX · 2023-08-31
> >
> > In the rebuttal, the authors have addressed my main concerns around evaluating the pipeline performance, post-processing metrics, table references, document diversity, downstream tasks, and related work. The additions of new supplementary tables and figures, expanded downstream evaluation, comparison with more OCR models, and a dedicated related work section demonstrate effort to improve the paper. While some limitations remain, the authors have clearly strengthened the manuscript in response to reviewer feedback.  I therefore increse my rating to Marginally above acceptance threshold.

---

### Official Review · Reviewer_z8qw · 2023-07-22
**Interesting paper with a very useful intention, however with shortcomings on pipeline availability, limited evaluation and explicitness of the requirements**

**Rating:** 6
**Confidence:** 4
**Clarity:** The paper is clearly written and prov…

**Strengths:**

- the paper has a very valuable intention and goal  - to provide a "large-scale dataset of historical American newspaper article texts and layouts that can be used for a wide variety of applications"
- the authors present an interesting overview of possible applications, ranging from "elucidating social science questions to training large language models and developing multimodal layout models to exploring world and family history."
- the authors' approach is constrained by cost constraints, which is a valuable contribution

**Additional Feedback:**

Seems this is still work in progress, it might be more useful to resubmit the paper when the project is finished.

**Correctness:**

The paper is in overall well written and easy to understand. There a number of discrepancies, e.g.
- Table 1 doesn't correspond to the numbers in the text and it is difficult to understand how the total numbers in the bottom row are calculated considering all the other rows are empty. The text talks about token counts and the table about bounding boxes;
- pipeline link https://github.com/dell-research-harvard/AmericanStories doesn't work
- the link to the dataset in the paper and the supplementary material are not the same

**Documentation:**

Appears proper

**Ethics:**

No ethical considerations

**Limitations:**

==== Evaluation =====
- the evaluation is done only on "ten scans consist of 597 content regions and 196,655 characters". Even though this is a considerable amount of character count to deal reliably with CER for OCR accuracy, it is critical to know what were the characteristics of these 10 scans. This is important to understand the validity of the evaluation. If the corpus is in the millions, it seems strange to pick 10 random documents for the evaluation, instead of carefully curating a set that would be representative of the different types of documents in the dataset

In addition, the https://github.com/dell-research-harvard/AmericanStories link doesn't work (it is not possible to verify the pipeline) and the authors claim comparison to commercial solution (only referred to an external reference), however there is no concrete summary of this in the paper itself in order to understand concretely the requirements definition.

**Opportunities For Improvement:**

Despite the very interesting topic and valuable intentions, the paper has a significant number of shortcomings with respect to the data description and pipeline availability, the evaluation of the approach and the explicitness of the requirements defined for its creation.

=== Pipeline availability and data description ===
- the https://github.com/dell-research-harvard/AmericanStories link doesn't work (it is not possible to verify the pipeline);
- the link in the paper used for the data points to dropbox and in the suplementary material is Huggingface
- The authors state 'The numbers reflect completed processing of around a third of Chronicling America scans, with processing of the full dataset to be completed before the author discussion period' --> this seems a bit preliminary  with only 1/3 of the scans done. In addition, the paper lacks clarity on what are the characteristics of this 1/3 of the scans - how representative they are for the whole collection, were they specifically selected, where they of a particular type, etc. This is critical to understand the validity of the results;
- Table 1 doesn't correspond to the numbers in the text and it is difficult to understand how the total numbers in the bottom row are calculated considering all the other rows are empty. The text talks about token counts and the table about bounding boxes;

==== Requirements ===
- the paper claims accuracy and cost objectives (however, only the costs were actually explicitly presented "a constrained cloud compute budget of $60,000 USD") there is no explicit mention of what the accuracy constraint. There is a reference to a short paper with overview of various methods - for the current paper to be self contained would be great if the authors extract this requirement from the reference into the paper.
- the authors claim a comparison to existing open source and commercial solutions - however no concrete reference to commercial solutions are given

**Relation To Prior Work:**

- authors refer to their own prior work, but overall related work is quite limited

**Summary And Contributions:**

The paper has a very valuable intention and goal  - to provide a "large-scale dataset of historical American newspaper article texts and layouts that can be used for a wide variety of applications". The authors present an interesting overview of possible applications, ranging from "elucidating social science questions to training large language models and developing multimodal layout models to exploring world and family history." In addition, the authors' approach is pragmatic - constrained by cost constraints, which is a valuable contribution. Despite the very interesting topic and valuable intentions, the paper has a significant number of shortcomings with respect to the data description and pipeline availability, the evaluation of the approach and the explicitness of the requirements defined for its creation.

---

> ### Author Response · Authors · 2023-08-22
> **Author Response**
>
> We thank reviewer x8qw for their thoughtful questions. We direct all reviewers to our updated submission, and respond individually here to the points raised in this review.
>
> *The https://github.com/dell-research-harvard/AmericanStories link doesn't work*
> Response: We apologize for this oversight; the link now works.
>
> *The link in the paper used for the data points to dropbox and in the suplementary material is Huggingface*
>
> Response: We have now switched this so that all links point to Huggingface.
>
> *Only 1/3rd of scans processed*
>
> Response: All scans in Chronicling America have now been processed and are available for download on Huggingface. The statistics about the dataset presented in the paper are now calculated on the full dataset, consisting of 1.14B content regions.
>
> *Confusion from Table 1*
>
> Response: We have clarified the table and ensured that all figures presented in the text match those in Table 1. Blank spaces in the table represent bounding boxes which were not OCRed (advertisements, photographs, etc), thus legibility was not predicted. In those cases, the bottom row represents the grand total detected content regions of that type.
>
> Bounding boxes reported in the table are OCRed if they are articles, headlines, image captions, or author bylines. The total number of tokens from all OCRed bounding boxes is reported in the text.
>
> *Comparison of OCR accuracy to existing solutions*
>
> Response: Supplementary Table 1 now compares the OCR accuracy and time/cost required for deployment of our pipeline on Chronicling America to a variety of other open-source and proprietary OCR engines. This table shows that EfficientOCR was the best performing architecture that met our cost requirements, as TrOCR would have been nearly 50 times as costly to deploy and Google Cloud Vision (GCV) would have cost even more. GCV makes significant layout errors when fed full newspaper page scans, which have complex layouts (Shen, 2021), and hence the line-level performance in Supplementary Table 1 cannot be replicated when it is fed scans. GCV charges per image, and the supplementary materials estimate a cost at current prices of 23 million USD to digitize LoCCA at the line image level, versus our 60K USD budget.
>
> *The evaluation is done only on "ten scans consist of 597 content regions and 196,655 characters"*
>
> Response: To evaluate the accuracy of the end-to-end pipeline, we needed to transcribe all content on the page, since the document layout analysis is applied at the page level. With an average of nearly 20K characters per scan, double entry with highly qualified undergraduate research assistants was very costly. We agree that a more diverse evaluation sample is important. Hence, we have two other samples on which we evaluate the OCR:
>
> 1. A day per decade sample, where we label 25 randomly selected lines of text per decade. We start with the 1850s, as the overwhelming share of content is post-1850 (less than 0.1% of content comes from the 18th century). These evaluations are presented in Supplementary Table 1, and we also give example line images for each decade in Supplementary Figure 1. OCR accuracy improves significantly over time, as the older scans are much more likely to have been poorly preserved or to have unusual visual signatures. Note that Figure 1 shows that the majority of the content comes from the later decades, where OCR quality is higher.
> 2. A randomly selected sample of 64 lines (1727 characters) from the period 1890-1920, that is taken from Carlson et al., (2023) and used to compare our pipeline to other open-source and proprietary OCR engines in Supplementary Table 1.

---

> > ### Comment · Reviewer_z8qw · 2023-08-29
> > **response to author's rebuttal**
> >
> > thank you for addressing the points in my review. Although the point of the evaluation has been addressed, i think this approach needs more creative thinking of how to be evaluated because of the current cost it requires and thus provides only a limited evaluation. I have increased my score to reflect the other changes

---

### Official Review · Reviewer_fRyu · 2023-07-26
**Great contribution to research community with open massive dataset of historical newspapaers and modular reusable data pipeline.**

**Rating:** 8
**Confidence:** 4

**Strengths:**

The main strengths of the submissions are directly stemming from its contributions. Massive dataset created with well-documented and modular pipeline are all ready to use by the broader research community. It will be invaluable to all sorts of social scientists.
Another strong suit is an explanation of balancing cost and quality under a limited budget by adapting deep learning techniques that are still accurate but less expensive to apply (and explaining the process of doing so). Researchers from institutions with less sizable funds will benefit from adapting described approach and building on top of it their own pipelines for other domains and languages.
Submission with primarily supplementary material proves beyond the shadow of a doubt the high quality of the research.
The ethical and social implications of a paper are positive. The research community will gain the ability to process at scale more digitized newspapers and conduct research on them. People learn from history, this work should be seen as an enabler for this learning process.

**Additional Feedback:**

1. It is unclear if manipulating the size of the word dictionary would be beneficial and how it influences the threshold of 0.82, but what might be worth exploring is adding prefixes (with - at the end) of the most common longer words that are at the highest risk of spanning two lines - if that makes sense.
2. It greatly contributes to the research community with no additional feedback.

**Clarity:**

The paper is very well written. Easy to follow, clearly structured with extensive supplementary material.

**Correctness:**

Yes, claims made in the sumbission are correct and dataset is self-contained, well-structured and thoriughly described.

**Documentation:**

Yes, good practices were followed. The dataset is hosted on HuggingFace https://huggingface.co/datasets/dell-research-harvard/AmericanStories under Creative Commons CC-BY license and supplementary scans can be downloaded from https://chroniclingamerica.loc.gov/
All details on how it was collected, how it’s maintained and organized are described in the paper. It’s very well documented.


**Ethics:**

No, I do not.

**Limitations:**

Distribution of newspaper articles in time (year) and space (states) as well as the language used, and topics journalists reported, are all used without filtering phrases that might be considered offensive by some or balancing dataset by applying stratified sampling by any feature. Authors processed what was available for them (all newspapers were already open to the public) and addressed and discussed all aspects openly. No feedback concerning it can be offered; everything is in order.

**Opportunities For Improvement:**

Opportunities for improvement are limited given effective funds allocation / spending and aiming at keeping quality at the highest possible level - recommending SOTA layout detection or OCR techniques that are more accurate but also more expensive is pointless, testing different architecture / tricks / quantization / distillation of models to make them run even cheaper is of course always an option.
However, what could be valuable for the research community is to have a more sizable gold dataset for each element of the pipeline and ready-to-run methods to benchmark them separately regarding both quality and resource consumption.
Raw annotations from each annotator are valuable too. What would be interesting is to study in more detail annotation discrepancies between multiple people - for example, for the illegibility of scans.


**Relation To Prior Work:**

Yes, how this work corresponds to prior work, including the Library of Congress’s Chronicling America project, is discussed in detail. What makes this work stand out are extra steps applied in the data processing pipeline to filter illegible scans, handle much better complex layouts and achieve higher OCR quality.
An example of prior work can be investigated here https://chroniclingamerica.loc.gov/lccn/sn86072160/1923-07-26/ed-1/seq-1/ocr/  - materials from this submission are of more rich and of higher quality.


**Summary And Contributions:**

The paper describes a novel data processing pipeline and how it is used to handle 20 million scans of pages from historical U.S. newspapers. All four steps of the pipeline: layout and line detection, legibility classification, word-level OCR, and article content association, are thoroughly discussed together with encountered challenges and an explanation of how they were addressed.
The main contribution on top of the modular and reusable pipeline using deep learning models is its outcome - a large-scale dataset of historical American newspaper article texts and layouts made publicly available for free and with a large variety of applications.

---

> ### Author Response · Authors · 2023-08-22
> **Author Response**
>
> We thank the reviewer for their thoughtful comments. We direct all reviewers to our updated submission, and respond individually here to the points raised in this review.
>
> *Gold Dataset for Pipeline Training and Evaluation*
>
> Response: We have released our layout, line detection, and word/character localization annotations, at https://huggingface.co/datasets/dell-research-harvard/AmericanStoriesTraining.
>
> *Annotation discrepancies*
>
> Response: In the context of legibility classification, inter-annotator agreement was 91% (n = 671 images). All but one of the disagreements were Legible/Questionable or Questionable/Illegible. Since our goal was to provide as much content as reasonably possible, we used the more legible label of the two when training our legibility classifier. A selection of annotator discrepancies is available as Figure 2 in the supplementary materials. The bounding box and OCR labels were completed by two highly skilled undergraduate research assistants and all discrepancies were resolved by hand. In these cases, the ground truth is easily verifiable and not a place where there is scope for any disagreement.
>
> *No “filtering phrases that might be considered offensive by some or balancing dataset by applying stratified sampling by any feature.”*
>
> Response: We understand why filtering or sampling is imperative for some research ends. However, our aim is to present the most complete possible version of the corpus. Social scientists and researchers in the digital humanities are part of our target audience, and often study content that some may consider offensive. If we selectively filtered content out, it could severely bias the conclusions of historical analyses of the texts, rendering them useless for researchers whose aim is to use them to understand historical contexts. Stratified sampling would  likewise destroy some features of the dataset (e.g., having a balanced panel for as many newspapers as possible) that are important for some research questions.  Researchers using the dataset can filter for offensive phrases, sample the corpus, or balance the dataset across space or time insofar as that benefits their research aims, and the licensing would allow them to share and redistribute these derivative datasets however they see fit.
>
> *The size of the word dictionary*
>
> Response: In developing our contrastive word-recognition model, we sought to balance a large vocabulary (which increases word recognitions and hence speed) with low misclassification (a larger vocabulary leads to more confusable words), as well as maintaining a reasonable model training time, which grows substantially in the size of the dictionary. We experimented with a variety of vocabulary sizes and found approximately 25,000 words worked well in practice to achieve a low CER, fast speed, and reasonable training time, but would welcome further experimentation with vocabulary size. The idea about adding words with dashes in the end is a great one - we plan to experiment more with expanding dictionary size based on these dataset-specific ideas as we move to digitize more historic datasets.
>
> The optimal word-recognition threshold (0.82 in our pipeline) does vary based on the dataset examined and vocabulary used. For example, in scans with less clear lettering, a lower threshold can be useful since individual letters are harder to recognize. Since determining the optimal threshold for a particular scan is not possible, we used the unified 0.82 for our processing.

---

### Author Response · Authors · 2023-08-22
**Global Response to Reviewers**

We thank all reviewers for their constructive feedback. We have responded to questions and concerns in individual responses and have posted a new version of the paper incorporating many of those suggestions. Additions include:

- **Updates to Reflect Complete Dataset.** As noted by reviewer z8qw, only ⅓ of the complete American Stories dataset was available as of our initial submission. We have now finished processing 100% of the dataset, and have updated the figures in Table 1 accordingly. The full dataset contains 1.14B content regions.
- **Added Literature Review.** Reviewers 1942, Z8NQ, and CqsX all requested additional discussion of related works. We added a section comparing our dataset to other similar collections, particularly Lee et. al. The Newspaper Navigator Dataset (2020).
- **Evaluation.** Several reviewers (Z8NQ, CqsX, and z8qw) asked for verification of American Stories’ OCR quality, particularly in comparison to alternative open source solutions. We present results comparing our chosen OCR framework with a variety of other open source and proprietary tools in Supplemental Table 1, as well as within the main text. We also expand evaluation beyond the ten fully annotated scans mentioned in the original paper.
- **Extended Applications.** We show additional potential dataset applications in Section 6, including topic classification, modeling content dissemination networks, and news story clustering.
- **Limitations.** We added Section 7 to the main paper, describing limitations and recommended uses of American Stories.
- **Training Data Release.** As suggested by reviewers Z8NQ and fRyu, we release all training images and annotations for greater reproducibility. This data can be found at https://huggingface.co/datasets/dell-research-harvard/AmericanStoriesTraining.
- **Additional Legibility and Textline Examples.** We added examples in Figure 2 and Supplemental Figure 1.

---

### Decision · Program_Chairs · 2023-09-22

**Decision:**

Accept (Poster)

**Comment:**

The paper received five reviews, three of them being on the positive side. The authors submitted rebuttals and some reviewers engaged in conversation with the authors, who clarified the raised weaknesses. In general, the reviewers appreciate the massive dataset created with well-documented and modular pipeline, the approach based on cost constraints. Overall, the reasons to accept the paper outweigh the negative points. However, the authors are asked to include the relevant points mentioned in the discussion to the camera ready.